METHODS AND RESOURCES

# Identification of substrates of palmitoyl protein thioesterase 1 highlights roles of depalmitoylation in disulfide bond formation and synaptic function

**Erica L. Gorenberg**[1,2☯‡], **Sofia Massaro Tieze**[1,2☯‡], **Betül Yücel**[1], **Helen R. Zhao**[1], **Vicky Chou**[1], **Gregory S. Wirak**[1], **Susumu Tomita**[3], **TuKiet T. Lam**[4,5], **Sreeganga S. Chandra**[1] *

**1** Departments of Neurology and Neuroscience, Yale University, New Haven, Connecticut, United States of America, **2** Interdepartmental Neuroscience Program, Yale University, New Haven, Connecticut, United States of America, **3** Departments of Neuroscience and of Cellular and Molecular Physiology, Yale University, New Haven, Connecticut, United States of America, **4** Departments of Molecular Biophysics and Biochemistry, Yale University, New Haven, Connecticut, United States of America, **5** Keck MS & Proteomics Resource, WM Keck Biotechnology Resource Laboratory, New Haven, Connecticut, United States of America

☯ These authors contributed equally to this work.
‡ These authors share first authorship on this work.
* sreeganga.chandra@yale.edu

**Data Availability Statement:** Data are available from ProteomeXchange in the PRIDE partner

## Abstract

Loss-of-function mutations in the depalmitoylating enzyme palmitoyl protein thioesterase 1 (PPT1) cause neuronal ceroid lipofuscinosis (NCL), a devastating neurodegenerative disease. The substrates of PPT1 are largely undescribed, posing a limitation on molecular dissection of disease mechanisms and therapeutic development. Here, we provide a resource identifying >100 novel PPT1 substrates. We utilized Acyl Resin-Assisted Capture (Acyl RAC) and mass spectrometry to identify proteins with increased *in vivo* palmitoylation in PPT1 knockout (KO) mouse brains. We then validated putative substrates through direct depalmitoylation with recombinant PPT1. This stringent screen elucidated diverse PPT1 substrates at the synapse, including channels and transporters, G-protein–associated molecules, endo/exocytic components, synaptic adhesion molecules, and mitochondrial proteins. Cysteine depalmitoylation sites in transmembrane PPT1 substrates frequently participate in disulfide bonds in the mature protein. We confirmed that depalmitoylation plays a role in disulfide bond formation in a tertiary screen analyzing posttranslational modifications (PTMs). Collectively, these data highlight the role of PPT1 in mediating synapse functions, implicate molecular pathways in the etiology of NCL and other neurodegenerative diseases, and advance our basic understanding of the purpose of depalmitoylation.

repository (project accession: PXD032052). Individual numerical source data, statistical analyses, and a list of PRIDE dataset files that underlie the summary data displayed in the figures are available in S1 Data.

**Funding:** This work was supported by the National Institutes of Health (R01 NS064963, S.S.C.; R01 NS110354, S.S.C.; R01 NS083846, S.S.C.; R21 NS094971, S.S.C.; R01 MH077939, S.T.; T32 NS007224, E.L.G. & S.M.T.; T32 NS041228, E.L.G. & S.M.T.; https://www.nih.gov), and the United States Department of Defense (W81XWH-17-1-0564, S.S.C.; https://www.defense.gov). The proteomic experiments were supported by the Yale/National Institute on Drug Abuse (NIDA) Neuroproteomic Center (NIH DA018343, S.S.C. & T.T.L.; https://medicine.yale.edu/keck/nida/general/mission/). The Q-Exactive Plus mass spectrometer was funded in part by National Institutes of Health Shared Instrumentation Grant from the Office of The Director, National Institutes of Health (S10OD018034, T.T.L; https://grants.nih.gov/grants/guide/pa-files/par-20-113.html). The funders had no role in study design, data collection and analysis, decision to publish, or preparation of the manuscript.

**Competing interests:** The authors have declared that no competing interests exist.

**Abbreviations:** ABE, Acyl-Biotin Exchange; ABHD, α/β-hydrolase domain-containing protein; Acyl RAC, Acyl Resin-Assisted Capture; ALS, amyotrophic lateral sclerosis; AMPAR, AMPA receptor; APT, acyl protein thioesterase; ASAH1, Acid Ceramidase; ATCC, American Type Culture Collection; CATD, Cathepsin D; dNEM, $d_5$-$N$-ethylmaleimide; HA, hydroxylamine; IACUC, Institutional Animal Care & Use Committee; ICC, immunocytochemistry; Ig, Immunoglobulin; IPA, Ingenuity Pathway Analysis; KO, knockout; LC–MS, MS, Liquid Chromatography–Mass Spectrometry; LFQ-MS, Label-Free Quantification Mass Spectrometry; LTP, long-term potentiation; MBP, myelin basic protein; MF, myelin fraction; NCL, neuronal ceroid lipofuscinosis; NEM, N-ethylmaleimide; PAT, palmitoyl acyl transferase; PFA, paraformaldehyde; PPT, palmitoyl protein thioesterase; PPT1, palmitoyl protein thioesterase 1; PTM, posttranslational modification; SPM, synaptic plasma membrane; TCEP, tris(2-carboxyethyl)phosphine; TEVC, two-electrode voltage clamp; TPP1, Tripeptidyl Peptidase 1; YARC, Yale Animal Resource Center; WT, wild-type.

## Introduction

Palmitoylation is the posttranslational addition of a C16:0 fatty acid chain to proteins, typically via thioester bond to cysteine residues. Palmitoylation is unique among lipid posttranslational modifications (PTMs) in its reversibility—other lipid PTMs typically last a protein's entire life span [1]. Palmitoyl groups are added by a family of 23 palmitoyl acyl transferases (PATs or ZDHHCs) [2,3] and are removed by distinct depalmitoylating enzymes—palmitoyl protein thioesterases (PPTs), acyl protein thioesterases (APTs), and α/β-hydrolase domain-containing proteins (ABHDs) [4–6]. The reversible nature of palmitate PTMs suggests that tight regulation of the palmitoylation/depalmitoylation cycle is necessary for proper protein function. Indeed, palmitoylation dynamics influence protein stability, function, membrane trafficking, and subcellular localization [7–10].

Systematic and unbiased proteomic studies of the mammalian brain have identified over 600 palmitoylated proteins, a high percentage of which are localized to synapses [11–13]. These large-scale screens suggest the importance of palmitoylation for synaptic function and neurotransmission. Studies have begun to identify the synaptic substrates of specific PATs [14–17]. However, it is still unclear for most palmitoylated proteins which enzyme facilitates depalmitoylation in the brain and at the synapse.

Loss-of-function mutations in the depalmitoylating enzyme palmitoyl protein thioesterase 1 (PPT1) lead to deficient depalmitoylation and synaptic function and result in the neurodegenerative disease NCL type 1 (*CLN1*) [6,18]. PPT1 therefore plays a critical role in the brain. However, the substrates of PPT1 are almost entirely unknown, with the exceptions of the presynaptic chaperone CSPα, the G-protein Goα, and the mitochondrial F1 ATP synthase subunit O [19–21], hindering the dissection of *CLN1* disease mechanisms. Identification of the repertoire of PPT1 substrates is needed to understand how depalmitoylation deficits impact neuronal health.

In this study, we therefore undertook a 2-step proteomic approach to identify PPT1 substrates and elucidate how depalmitoylation contributes to synaptic function and phenotypes of NCL. Our screen greatly expands the repertoire of known palmitoylated proteins in the brain. We detected PPT1 in all synaptic subcompartments and found high levels of PPT1 enzyme activity in the synaptic cytosol. We identified approximately 10% of palmitoylated synaptic proteins as PPT1 substrates. These proteins display increased palmitoylation independently of protein expression *in vivo*. The validated substrates fall into 9 classes, which, strikingly, are related to phenotypes observed in PPT1 knockout (KO) mice and *CLN1* patients, including seizures, decreased synapse density, mitochondrial dysfunction, synaptic vesicle endocytic deficits, impaired long-term potentiation (LTP), and retinal degeneration [22–27]. Notably, PPT1 depalmitoylation sites on validated substrates are frequently cysteine residues that participate in disulfide bonds, suggesting that a novel function of palmitoylation may be to mediate these interactions. Our classification of PPT1 substrates provides a resource to enhance our understanding of depalmitoylation and its contributions to synaptic function, neuronal health, and the molecular basis of NCL.

## Results

### WT and PPT1 KO brain palmitome expands the repertoire of known palmitoylated proteins in the brain and draws links between NCLs

Neuronal substrates of PPT1 are expected to exhibit increased palmitoylation in PPT1 KO mouse brains, but it is unclear whether these changes are independent of protein levels. Therefore, we designed an experimental workflow to analyze both the proteome and palmitome

from the same wild-type (WT) and PPT1 KO brains by Label-Free Quantification Mass Spectrometry (LFQ-MS; **Fig 1A**). This scheme allowed for comparisons across both genotypes (WT versus PPT1 KO) and "-omes" (proteome versus palmitome) and utilized *in vivo* palmitoylation changes to identify PPT1 substrates. Synaptosomes and whole brain homogenates from WT and PPT1 KO mice were processed for Acyl Resin-Assisted Capture (Acyl RAC), a method to selectively capture endogenously palmitoylated proteins, as previously described [28,29]. Briefly, disulfide bonds were reduced with tris(2-carboxyethyl)phosphine (TCEP), and resulting free thiols were blocked with N-ethylmaleimide (NEM; **Fig 1Bi**). Palmitate groups were removed with hydroxylamine (HA; **Fig 1Bii**) and resulting unblocked free thiols were bound to thiopropyl sepharose beads (**Fig 1Biii**). Bound proteins were then eluted for Liquid Chromatography–Mass Spectrometry (LC–MS/MS; **Fig 1Biv**), selectively isolating previously palmitoylated proteins (**Fig 1C and 1D**). Proteome fractions were collected prior to depalmitoylation by HA (**Fig 1A and 1Bii**), while palmitome fractions were isolated following pull-down on and elution from thiopropyl sepharose beads (**Fig 1Biv**). Both the proteome and palmitome were analyzed by LFQ-MS following digestion and alkylation with iodoacetamide to modify previously palmitoylated cysteines with a carbamidomethyl moiety (**Fig 1Bv–vii**).

The Acyl RAC workflow was conducted on WT and PPT1 KO mouse brains ($n$ = 3, each in technical triplicate; age = 2 months) in a pairwise fashion. We examined proteins common to all 3 experiments, cognizant of the fact that this stringency is likely to result in underestimation of the number of PPT1 substrates. The resulting MS analysis of the palmitome identified 1,795 common palmitoylated proteins (**S1A and S1B Fig**), and analysis of the proteome identified 1,873 common proteins (**S1C and S1D Fig**) between genotypes. A limitation of Acyl RAC is the inability to distinguish palmitoylation from other lipid adducts with thioester linkages to cysteine [30]. Therefore, we compared our brain palmitome to a database of palmitoylated proteins identified by Acyl RAC, Acyl-Biotin Exchange (ABE), or CLICK chemistry studies. Most of our hits (69.8%; **S1A and S1B Fig**) were validated by this repository [31], which strengthens our confidence in the successful identification of palmitoylated proteins by Acyl RAC. Our palmitome expands the number of previously known palmitoylated proteins in the brain [11–13].

To highlight meaningful palmitome and proteome changes and normalize for low-level nonspecific changes such as auto-acylation and auto-depalmitoylation, we visualized the data as volcano plots comparing PPT1 KO/WT (**S1B Fig**). Only 12 palmitoylated proteins display significantly increased expression in PPT1 KO brains (**S1B Fig**). Surprisingly, PPT1 is palmitoylated, but its levels are significantly decreased in the PPT1 KO (**S1B Fig**); detection of PPT1 may be explained by sequence similarity with PPT2 or translation of a truncated, catalytically inactive N-terminal PPT1 transcript [22].

We inspected the status of other known depalmitoylating enzymes—PPT2, APTs, and ABHDs—and found their levels unaltered, suggesting that they do not display compensatory up-regulation of protein expression due to loss of PPT1 (ABHD12, APT1, and APT2; **S1D Fig**, green points). We find increased levels of Acid Ceramidase (ASAH1), Cathepsin D (CATD), Lysosome Membrane Protein 2 (SCRB2), Prosaposin (SAP), and Tripeptidyl Peptidase 1 (TPP1) in the PPT1 KO palmitome (**S1B Fig**) and proteome (**S1D Fig**), while the remaining proteins with elevated palmitoylation—C1QC, PTTG, CATF, AP1B1, SRBS1, SRBS2, and RTN3—do not display concurrent elevation of protein expression. These 5 proteins (ASAH1, CATD, SCRB2, SAP, and TPP1) were previously shown to be increased in PPT1 KO or *CLN1* models [32–34]. The up-regulation of these proteins, occuring here prior to symptom onset and substantial neuron loss, remains a hallmark at later disease time points in the brain (4 months) [35]. Intriguingly, TPP1 and CATD loss-of-function mutations cause other forms of NCL (*CLN2* and *CLN10*, respectively) and isoforms of SAP accrue in *CLN1* brains [34],

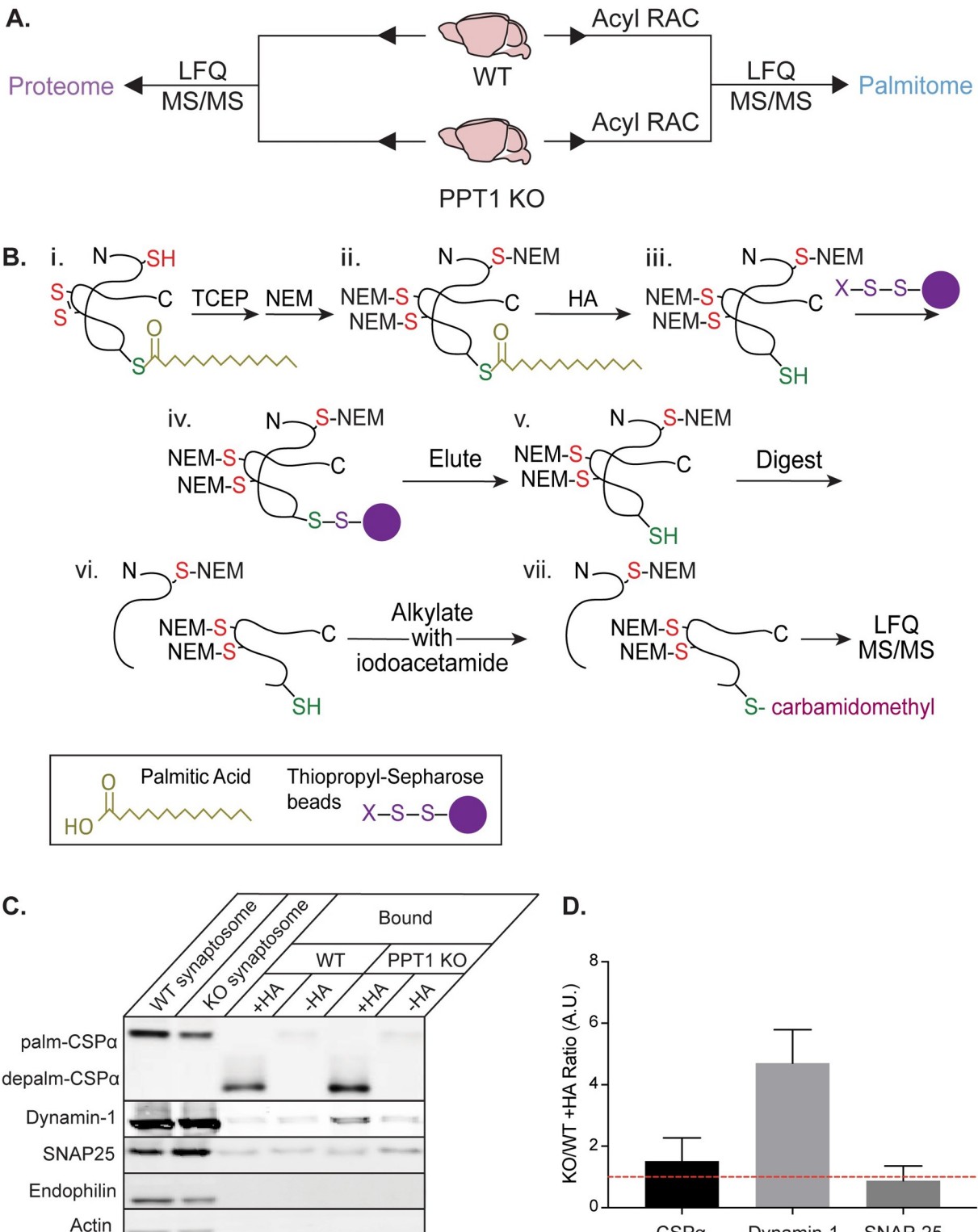

**Fig 1. Generation of the proteome and palmitome from WT and PPT1 KO brains. (A)** Proteins were isolated from WT and PPT1 KO whole brains or synaptosomes for proteomic analysis by LFQ-MS to identify overall protein expression (proteome; left) or following Acyl RAC to identify palmitoylated proteins (palmitome; right). **(B)** Schematic of Acyl RAC. (i) Disulfide bonds were reduced with TCEP and residual thiol groups were blocked with NEM. (ii) Palmitate groups were hydrolyzed with HA, leaving behind an unblocked free thiol. (iii) Free thiols were bound to thiopropyl sepharose beads, allowing for isolation of palmitoylated proteins. These proteins were then (iv) eluted from beads

with DTT, (v) digested, and (vi) alkylated with iodoacetamide, followed by (vii) LFQ-MS/MS analysis. **(C)** Pull-down and immunoblotting of palmitoylated proteins. Addition of HA allowed CSPα, dynamin-1, and SNAP-25, all of which are palmitoylated proteins, to be pulled down on thiopropyl sepharose beads, but not endophilin or actin, which are not palmitoylated. Note that addition of HA removes palmitate and causes a molecular weight shift of palmitoylated CSPα to depalmitoylated CSPα. **(D)** Quantification of western blot +HA band intensity (A.U), plotted as a ratio of KO/WT (**S1 Data**). CSPα is a known PPT1 substrate and shows elevated palmitoylation in KO/WT, while SNAP-25 does not (*n* = 3 to 9 experiments). Acyl RAC, Acyl Resin-Assisted Capture; HA, hydroxylamine; KO, knockout; LFQ-MS, Label-Free Quantification Mass Spectrometry; NEM, N-ethylmaleimide; PPT1, palmitoyl protein thioesterase 1; TCEP, tris(2-carboxyethyl)phosphine; WT, wild-type.

suggesting a common etiological pathway among NCLs. ASAH1 and TPP1 also accumulate in the spinal cord of PPT1 mice at 3 months of age and remain elevated at 7 months, suggesting an important pathological role for these proteins across the central nervous system [36].

Palmitoylated protein levels in the brain palmitome are highly correlated across genotypes (m = 0.9753 ± 0.0030; $R^2$ = 0.9857; **S1E Fig**). Similarly, protein expression levels in the brain proteome were also highly correlated between WT and PPT1 KO brains (m = 0.9802 ± 0.0023; $R^2$ = 0.9916; **S1F Fig**), in line with the lack of overt differences in neurological phenotypes at this age (2 months) [22]. Overall, 72% of palmitoylated proteins (1,290/1,795) were also identified in the proteome (**S1G Fig**), allowing us to distinguish increased palmitoylation from increased protein expression. For proteins with significantly increased expression in the palmitome (**S1B Fig**), we plotted proteome versus palmitome expression levels in PPT1 KO after normalization to WT (**S1H Fig**). In the case of the lysosomal proteins ASAH1, CATD, SCRB2, SAP, and TPP1 (**S1H Fig**), the proteome to palmitome ratios were unchanged or decreased, precluding us from defining these 5 proteins as substrates. Overall, there were very few significant changes in the whole brain palmitome (**S1B Fig**).

## Putative PPT1 substrates are enriched at the synapse

Immunocytochemistry (ICC) demonstrates that PPT1 is detected at synapses when expressed in neurons (**S2A Fig**), consistent with previous literature detailing the presence of PPT1 in axons, synaptosomes, and synaptic vesicles in addition to its expected lysosomal localization [23,37,38]. We confirmed the presence of endogenous PPT1 at the synapse by immunoblotting subcellular fractionation samples generated from WT and PPT1 KO mouse brains (age = 2 months). PPT1 is detected in all synaptic fractions in WT and is absent in PPT1 KO (**S2B Fig**), while markers of subcellular fraction purity are appropriately enriched for both genotypes (**S2C Fig**). Furthermore, PPT1 enzymatic activity correlated with protein expression in WT synaptosomes and subcellular fractions and was particularly enriched in the synaptic cytosol (**S2C and S2D Fig**).

As we identified very few palmitoylation changes between WT and PPT1 KO from the whole brain analyses, including changes to known substrates of PPT1—CSPα, Goα, and F1 ATPase subunit O [19–21]—we reasoned that subcompartmentalization of PPT1 may have diluted the LFQ signal. Therefore, we proceeded to isolate synaptosomes from brains of PPT1 KO and WT littermates to enrich for PPT1 and its potential synaptic substrates. We then subjected synaptosomes to the same Acyl RAC workflow (**Fig 1B**). We identified high degrees of overlap between the 3 synaptosomal preparations for the palmitome (1,379 common proteins; **Fig 2A and 2B**) and the proteome (1,826 common proteins; **Fig 2C and 2D**), indicating excellent reproducibility.

When we compared synaptic proteomes across genotypes, we corroborated that protein expression between WT and PPT1 KO is highly correlated (m = 0.969 ± 0.0016; $R^2$ = 0.9666; **Fig 2F**). Consistent with a synaptic localization (**S2 Fig**) and previous literature, we identified PPT1 in the synaptosomal proteome (**S1 Table**) [23,37,38]. Notably, PPT1 is the only protein with significantly decreased levels in PPT1 KO synaptosomes, and only 10 proteins exhibited

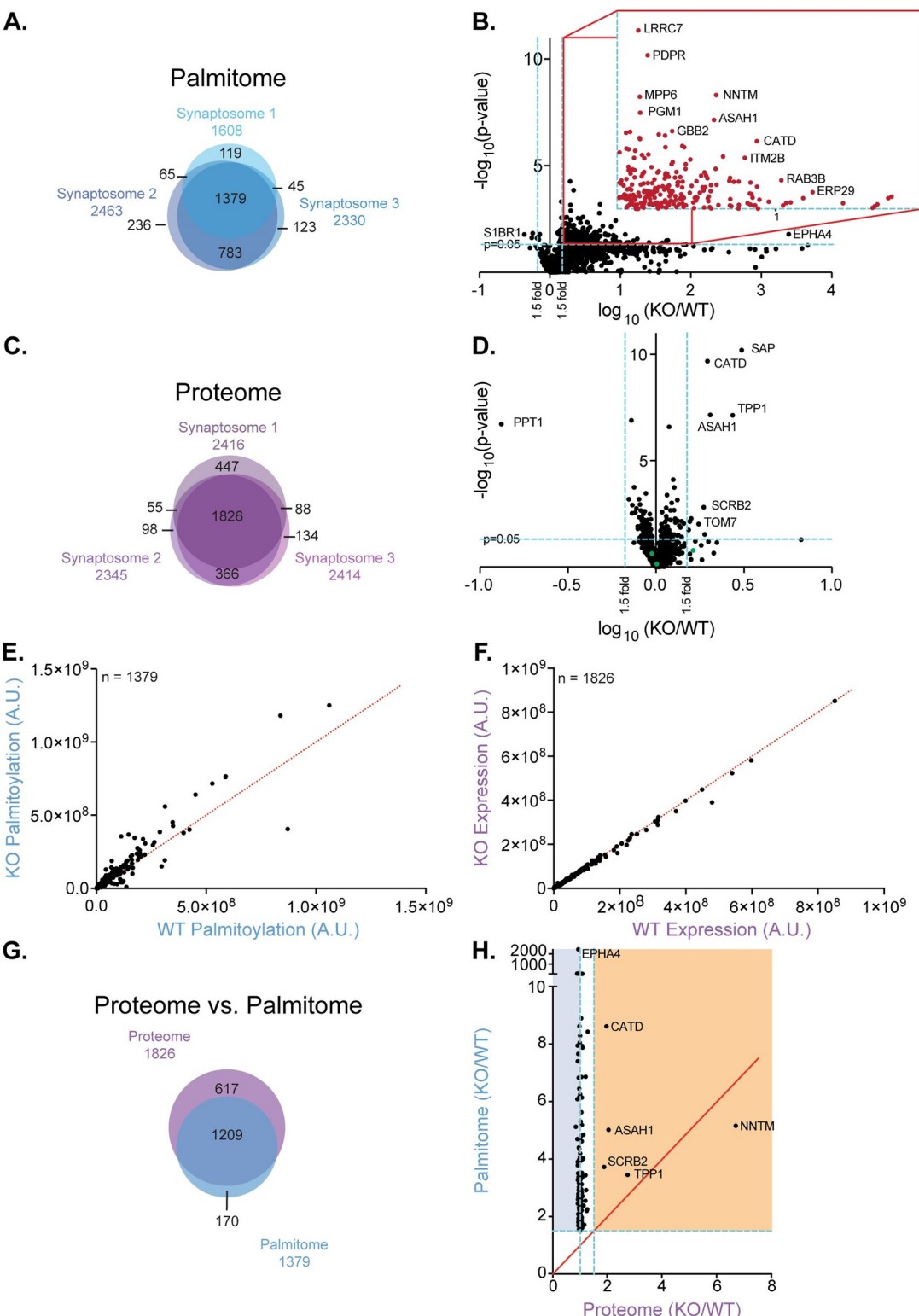

**Fig 2. Putative PPT1 substrates are enriched at the synapse and display persistent palmitoylation. (A)** Venn diagram of 3 palmitome experiments exhibits 1,379 common proteins. **(B)** Volcano plot indicates 242 proteins are significantly differentially palmitoylated (4 decrease; 238 increase; 1.5-fold, $p < 0.05$; blue lines; **S2 Table, S1 Data**). Inset shows significantly increased proteins. The $p$-value was calculated using a 2-tailed $t$ test on 3 biological and 3 technical replicates. While the technical replicates do not meet the $t$ test condition for independence, we chose to proceed in this manner to generate more putative

substrates that could later be validated with higher stringency. **(C)** Venn diagram of 3 independent proteome experiments exhibits 1,826 common proteins. **(D)** Volcano plot of fold change between genotypes (PPT1 KO/WT littermates) for proteins identified in all 3 biological replicates. A total of 11 proteins are significantly differentially expressed (1 decrease (PPT1); 10 increase; 1.5-fold, $p < 0.05$; blue lines; **S1 Data**). Other depalmitoylating enzymes (green points) do not display significant compensatory up-regulation of protein expression. **(E)** Palmitoylated protein expression is somewhat correlated between genotypes ($R^2 = 0.899$; m = 1.134 ± 0.0197), with increased palmitoylation in PPT1 KO (**S1 Data**). **(F)** Protein expression is almost perfectly correlated between genotypes ($R^2 = 0.9966$; m = 0.969 ± 0.0016; **S1 Data**). Red lines indicate 1:1 WT to PPT1 KO protein expression ratio. **(G)** Venn diagram of 1,209 common proteins between whole brain proteome and palmitome ($n = 3$ each). **(H)** Protein expression levels compared to palmitoylation levels for significantly changed proteins in palmitome. Proteins in orange region are increased (1.5-fold, $p < 0.05$) in both proteome and palmitome. Proteins in blue region display decreased or unchanged protein expression and increased palmitoylation. Red line indicates equal expression and palmitoylation levels ($x = y$). While NNTM has a high fold change, it does not meet the $p$-value criterion for significantly increased protein expression ($p = 0.0522$) and is therefore excluded from this category (**S1 Data**). ASAH1, Acid Ceramidase; CATD, Cathepsin D; KO, knockout; PPT1, palmitoyl protein thioesterase 1; TPP1, Tripeptidyl Peptidase 1; WT, wild-type.

significantly increased expression (proteome KO/WT ratio >1.5; $p < 0.05$; **Fig 2D**). The proteins with increased expression were also identified in the whole brain analysis (**S1 Fig**), with the exceptions of the mitochondrial proteins TOM7, CX7A2, and NDUF4. As with the whole brain, we also identified other depalmitoylating enzymes in the synaptic proteome and found their levels to be unaltered (ABHD12, ABHDA, ABHGA, and APT2; **Fig 2D**, green points).

Notably, there was a much lower correlation between WT and PPT1 KO palmitomes (m = 1.134 ± 0.0197; $R^2 = 0.8986$; **Fig 2E**), with many (610; 44.3%) points above the $y = x$ unity line indicating elevated palmitoylation in PPT1 KO synaptosomes. The remarkable increase in palmitoylation in PPT1 KO synaptosomes (**Fig 2B**) compared to whole brain (**S1B Fig**), supports local PPT1-mediated depalmitoylation of synaptic substrates, as a depalmitoylation deficiency is expected to lead to an accumulation of palmitoylated targets.

Indeed, the proteins of greatest interest as potential PPT1 substrates were those with significantly increased palmitoylation in PPT1 KO compared to WT ($p < 0.05$, 1.5-fold change cutoff). We identified 227 such proteins (**Fig 2B**, **S2 Table**) that were considered putative PPT1 substrates. These include the previously known substrates, CSPα, F1 ATP synthase subunit O, and Goα [19–21]. Most of the putative substrates were also identified in the whole brain proteome (77.1%; blue points; **S1D Fig**) and palmitome (72.0%; blue points; **S1B Fig**), but their palmitoylation levels were not significantly altered, likely due to the synaptic compartmentalization of PPT1 (**S2 Fig**).

Approximately 66% of proteins present in the synaptic palmitome were also identified in the synaptic proteome ($n = 1,209$; **Fig 2G**). To confirm that the increased levels of putative PPT1 substrates in the palmitome are not due to elevated protein expression, we plotted the KO/WT ratio to elucidate the relationship between palmitoylation and protein levels (**Fig 2H**). In line with our initial analyses (**Fig 2B** and **2D**), we found only 4 proteins with increased expression and increased palmitoylation in PPT1 KO synaptosomes: ASAH1, CATD, SCRB2, and TPP1 (**Fig 2H**; orange region). For this group of proteins, we could not adequately discriminate increases in palmitoylation resulting from increased protein expression versus PPT1 deficiency. Hence, these proteins may exhibit depalmitoylation-dependent degradation or may not be PPT1 substrates. Therefore, these 4 proteins were removed from subsequent analyses, along with 19 additional proteins which were not identified in the proteome, as we could not confirm whether their increased palmitoylation was independent of protein expression level (**S2 Table**).

The remaining putative substrates identified in both proteome and palmitome ($n = 204$; blue/white region, **Fig 2H**) showed elevated palmitoylation without a commensurate increase in protein levels. Significantly, we identified the putative substrate dynamin-1 in this subset, which showed increased Acyl RAC pulldown in the PPT1 KO synaptosomes by western

blotting (**Fig 1C and 1D**) unlike the control SNAP-25, congruent with these results. The relative increase in palmitoylation in PPT1 KO ranged from 1.5-fold at the lower limit to 2400-fold (Ephrin A4; EphA4) at the upper limit, with known PPT1 substrates CSPα (DNJC5; 5.12-fold), F1 ATPase subunit O (ATPO; 3.39-fold), and the Goα subunit (GNAO; 4.81-fold) at 3- to 5-fold (**Fig 2H**).

Finally, we examined the remaining putative PPT1 substrates biochemically for evidence of palmitoylation. By analyzing individual peptides identified by LFQ-MS for each protein, we identified cysteine-containing peptides that were originally palmitoylated by their now derivatized carbamidomethyl moiety. Out of the 204 putative substrates, we detected a specific palmitoylation site for 101 proteins (49.5%; **S3 Table**).

## Direct depalmitoylation with PPT1 validates putative synaptic substrates

To validate the 204 putative PPT1 substrates identified in our initial Acyl RAC screen, we developed a modified secondary Acyl RAC screen in which recombinant mouse PPT1 expressed in HEK293T cells is used as the depalmitoylating reagent, rather than HA. We previously established that recombinant PPT1 can depalmitoylate its substrate CSPα *in vitro* [29]. Therefore, mouse PPT1 was incubated with WT synaptosomes at Step ii of the Acyl RAC protocol (**Figs 1Bii and 3Ai**).

MS analysis of PPT1-depalmitoylated samples identified 971 common proteins over 3 biological replicates (**Fig 3Aii**). A total of 138 of the 204 putative substrates identified in the primary screen were validated (**Fig 3B**). Significantly, investigation of the peptide data allowed us to identify the carbamidomethyl-modified peptide and thus the specific palmitoylated cysteine residue, for a subset of these proteins, as described above. We used this information to define degrees of stringency for the categorization of protein hits from the validation screen, as follows: (1) High-confidence PPT1 substrates are those for which matching carbamidomethyl peptides were identified in both the initial Acyl RAC WT/PPT1 KO screen and the PPT1-mediated depalmitoylation validation screen ($n = 26$; **Table 1**). (2) Medium-confidence substrates are those identified in both the primary and validation screens with $\geq 2$ unique peptides per protein and a confidence score $>100$ ($n = 112$; **Table 2**). (3) Residuals were proteins identified as putative substrates only in the secondary validation screen ($n = 831$; **S4 Table**).

To determine whether our validated substrates are corroborated by prior computational or experimental evidence of palmitoylation, we cross-referenced validated hits with CSS Palm [39], Swiss Palm [31], and an ABE capture study [12]. Notably, these comparisons identified evidence of palmitoylation for 100% (26/26) of high-confidence substrates (**Table 1**) and 94% (105/112) of medium-confidence substrates (**Table 2**). Validated proteins also include all known PPT1 substrates—CSPα, Goα, and F1 subunit O of mitochondrial ATP synthase—and exclude substrates of other depalmitoylating enzymes, such as PSD-95, an abundant synaptic protein that is exclusively depalmitoylated by ABHD17A, 17B, and 17C [40]. Putative APT1 substrates such as RAB9A, RAB6A, N-RAS, R-RAS, GAP-43, and SNAP-23 are also excluded [20,41,42]. We also screened our results against a PPT1 interactome dataset ($n = 64$) [24] and found 4 high-confidence substrates (DYL2, LDHB, STXB1, and THY1) and 4 medium-confidence substrates (DYN1, GBB2, TERA, and VA0D1) to be present. Comparison with another PPT1 interactome dataset ($n = 852$) [43,44] validated 7 high-confidence substrates (AT1A1, AT1B1, ATPO, DYL2, LDHB, NDKA, and VDAC2) and 22 medium-confidence substrates (1433B, ACTN1, ADDG, AIFM1, ATPG, CISY, CLH1, CLUS, COX2, CRIP2, EFTU, ERP29, FAS, HSP74, HXK1, KIF5C, M2OM, ODO1, ODPX, PGM1, PRDX6, and TERA). Together, these data comprehensively substantiate that validated PPT1 substrates are palmitoylated proteins and that PPT1 has substrate specificity.

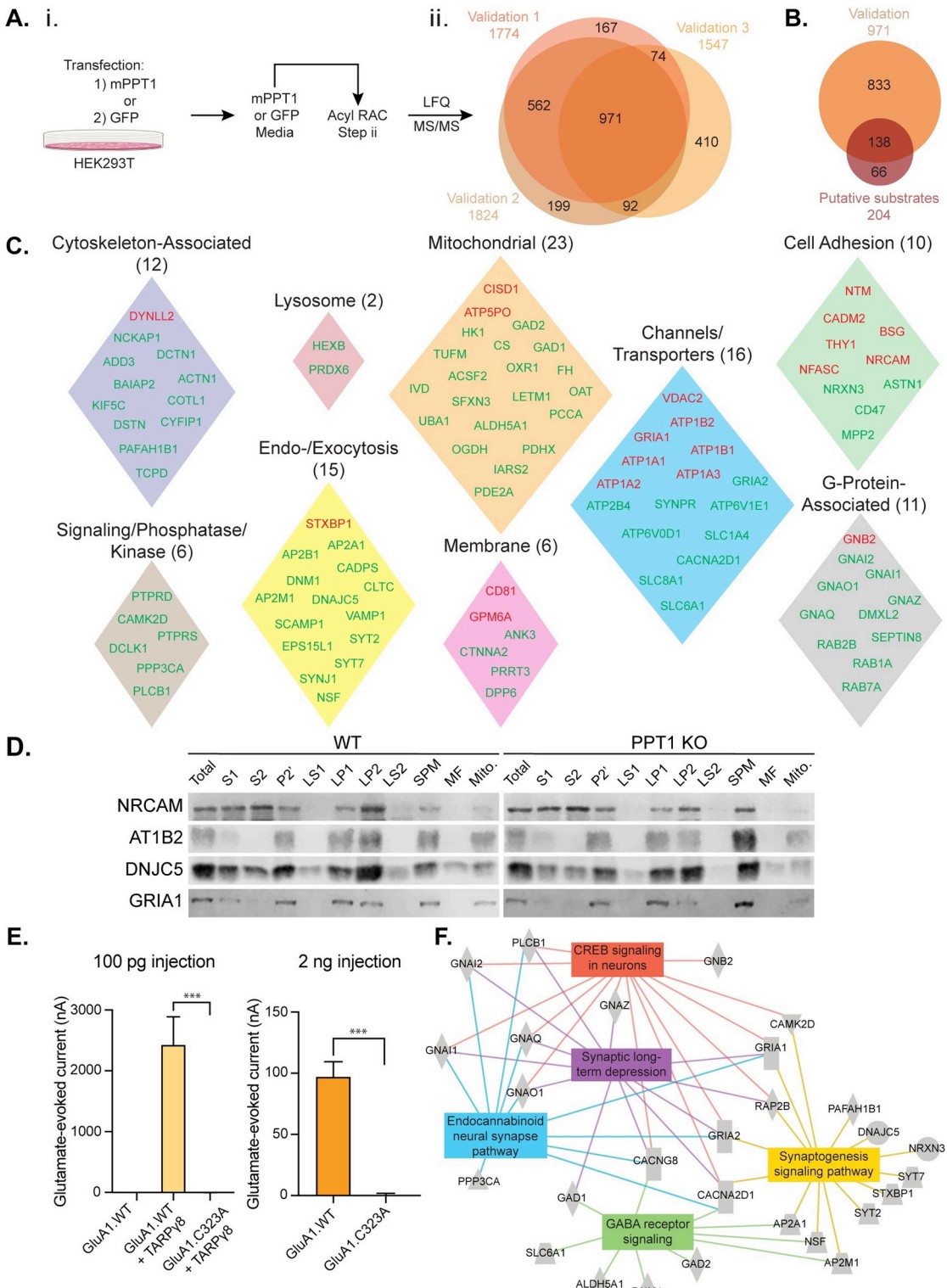

**Fig 3. Direct depalmitoylation with PPT1 validates putative synaptic substrates. (A)** (i) Schematic of PPT1-mediated depalmitoylation assay, in which solubilized synaptosomes are treated with recombinant mPPT1 or GFP generated from HEK293T cells during Acyl RAC (**Fig 1Bii**). (ii) Venn diagram of PPT1-mediated depalmitoylation assay (*n* = 3 biological replicates, *n* = 3 technical replicates) shows 975 proteins common to replicates. **(B)** Comparison of PPT1-mediated depalmitoylation proteins to putative substrates results in 138 validated PPT1 substrates. **(C)** The 138 combined high- (red) and medium- (green) confidence

validated proteins fall into curated UniProt functional and subcellular location groupings (UniProt gene names are indicated). **(D)** Western blots of subcellular fractionation samples prepared from whole brains of WT and PPT1 KO mice. **(E)** *Xenopus laevis* oocytes were injected with WT GluA1 (GluA1.WT) or C323A mutant GluA1 cRNA (100 pg) alone or with auxiliary factor TARPγ8 (100 pg). Glutamate (5 μm)-evoked current with cyclothiazide (50 μM) was measured by 2-electrode voltage-clamp recording ($n = 4$; *** $p < 0.001$). GluA1.WT (2 ng) or GluA1.C323 (2 ng) were injected at higher concentrations to test spontaneous channel formation in the absence of auxiliary factors and glutamate (1 mM)-evoked current with cyclothiazide (50 μm) was recorded ($n = 6$; *** $p < 0.001$). Data are presented as mean ± SEM (**S1 Data**). **(F)** IPA displays enrichment of high- and medium-confidence substrates for synaptic long-term depression, synaptogenesis signaling, GABA receptor signaling, endocannabinoid neural synapses, and CREB signaling in neurons. Acyl RAC, Acyl Resin-Assisted Capture; IPA, Ingenuity Pathway Analysis; KO, knockout; PPT1, palmitoyl protein thioesterase 1; WT, wild-type.

## Classification of PPT1 substrates highlights diverse synaptic functions

To obtain a better understanding of the functional role of PPT1-dependent depalmitoylation, the 138 high- and medium-confidence PPT1 substrates were analyzed using UniProt Gene Ontology, resulting in 9 distinct classes (**Fig 3C**): (1) cytoskeletal proteins ($n = 12$); (2) lysosomal proteins ($n = 2$); (3) mitochondrial proteins ($n = 23$); (4) synaptic cell adhesion molecules ($n = 10$); (5) channels and transporters ($n = 16$); (6) kinases and phosphatases ($n = 6$); (7) exo- and endocytic proteins ($n = 15$); (8) membrane proteins ($n = 6$); (9) G-protein–associated molecules ($n = 11$); and (10) other proteins ($n = 37$). A total of 75% of high- and medium-confidence validated proteins (104/138) fall nicely into defined classes with functions at the synapse. Our results agree with prior evidence that palmitoylation regulates ion channels and transporters, G-protein–associated molecules, and mitochondrial proteins [45–47]. Our findings that synaptic adhesion molecules and endocytic proteins undergo PPT1-mediated depalmitoylation are novel.

To gain insight into the role of PPT1-mediated depalmitoylation at the synapse, we searched for the subsynaptic localization of high-confidence substrates described in prior studies. Most proteins were demonstrated to be localized to the synaptic plasma membrane (SPM; 15/26), with the remaining proteins found in the synaptic cytosol (6/26) or at organelles such as mitochondria and lysosomes (3/26; **S5 Table**; UniProt). None of these substrates appear to be localized exclusively to the postsynaptic cytosol. The existence of both SPM and cytosol substrates is intriguing and in line with the subcellular localization of PPT1 (**S2B Fig**). Western blot analysis of subcellular fractions from WT and PPT1 KO brains demonstrates that select validated PPT1 substrates (NRCAM, AT1B2 (ATP1B2), DNJC5 (CSPα), and GRIA1 (GluA1); **Fig 3D**) and synaptic markers (**S2B Fig**) are localized to the expected synaptic subcellular fraction.

We next investigated peptides with carbamidomethyl-modified cysteines (known palmitoylation sites) detected for high-confidence PPT1 substrates to identify common features. Interestingly, we find that the carbamidomethyl peptides for most membrane-associated substrates are extracellular and face the synaptic cleft (**S5 Table**). Strikingly, we find that for all 12 peptides identified as extracellular and carbamidomethyl-modified, the modified cysteines participate in disulfide bonds in the mature protein (**S5 Table**). This suggests that cysteines could be shared for palmitoylation and disulfide bond formation, and perhaps that intracellular palmitoylation precedes disulfide bond formation, as proposed previously [48–50]. Many of the proteins containing these extracellular peptides fall into the broad classes of cell adhesion molecules (NFASC, NRCAM, CADM2/SynCAM2, NTRI, THY1, and BASI), and channels/transporters (GRIA1, AT1B1, and AT1B2; **Fig 3C**).

To investigate the importance of cysteines that are shared for palmitoylation and disulfide bond formation, we studied the AMPA receptor (AMPAR) subunit GluA1 (GRIA1). GluA1 was identified as a high-confidence PPT1 substrate with a palmitoylation site on C323. Based on a recent crystal structure, C323, which typically forms a disulfide bond to C75 (**S5 Table**), is found in an extracellular loop [51]. This site has not been previously identified as

**Table 1. High-confidence validated PPT1 substrates.**

| Protein name (UniProt ID) | Gene name | Description | Palmitome ratio | Palmitome p-value | Proteome ratio | Proteome p-value | CRAPome (%) | Kang and colleagues | Swiss Palm (%) | CSS Palm |
|---|---|---|---|---|---|---|---|---|---|---|
| AT1A1_MOUSE | Atp1a1 | Sodium/potassium-transporting ATPase subunit alpha-1 | 1.831 | 0.007 | 0.960 | 0.318 | 10.949 | | 5.882 | X |
| AT1A2_MOUSE | Atp1a2 | Sodium/potassium-transporting ATPase subunit alpha-2 | 1.744 | 0.003 | 1.068 | 0.498 | 22.871 | | 5.882 | |
| AT1A3_MOUSE | Atp1a3 | Sodium/potassium-transporting ATPase subunit alpha-3 | 1.746 | 0.030 | 1.042 | 0.695 | 24.574 | | 5.882 | |
| AT1B1_MOUSE | Atp1b1 | Sodium/potassium-transporting ATPase subunit beta-1 | 2.180 | 0.019 | 0.982 | 0.536 | 0.243 | X | 5.882 | |
| AT1B2_MOUSE | Atp1b2 | Sodium/potassium-transporting ATPase subunit beta-2 | 6.852 | 0.018 | 1.043 | 0.471 | 0.000 | X | 5.882 | X |
| ATPO_MOUSE | Atp5po | ATP synthase subunit O, mitochondrial | 6.138 | 0.044 | 1.060 | 0.277 | 0.000 | X | 5.882 | |
| BASI_MOUSE | Bsg | Basigin | 2.695 | 0.016 | 0.927 | 0.101 | 9.732 | X | 5.882 | X |
| CADM2_MOUSE | Cadm2 | Cell adhesion molecule 2 | 1.975 | 0.004 | 0.988 | 0.559 | 0.000 | | 11.76 | X |
| CD81_MOUSE | Cd81 | CD81 antigen | 2.554 | 0.042 | 1.035 | 0.940 | 0.243 | | 29.41 | |
| CISD1_MOUSE | Cisd1 | CDGSH iron-sulfur domain-containing protein 1 | 1.920 | 0.050 | 0.992 | 0.635 | 1.703 | | 35.29 | |
| DYL2_MOUSE | Dynll2 | Dynein light chain 2, cytoplasmic | 4.192 | 0.027 | 0.996 | 0.808 | 9.246 | | 0 | X |
| GBB2_MOUSE | Gnb2 | Guanine nucleotide-binding protein G(I)/G(S)/G(T) subunit beta-2 | 2.963 | 0.003 | 0.985 | 0.636 | 15.329 | | 2.92 | X |
| GPM6A_MOUSE | Gpm6a | Neuronal membrane glycoprotein M6-a | 5.625 | 0.007 | 1.043 | 0.658 | 0.243 | X | 5.882 | X |
| GRIA1_MOUSE | Gria1 | Glutamate receptor 1 | 1.866 | 0.046 | 0.992 | 0.635 | 0.000 | X | 5.882 | X |
| KCRB_MOUSE | Ckb | Creatine kinase B-type | 1.815 | 0.029 | 0.959 | 0.242 | 39.659 | | 11.76 | |
| LDHB_MOUSE | Ldhb | L-lactate dehydrogenase B chain | 1.928 | 0.050 | 1.006 | 0.939 | 39.903 | X | 11.76 | |
| LGI1_MOUSE | Lgi1 | Leucine-rich glioma-inactivated protein 1 | 2.047 | 0.013 | 1.014 | 0.883 | 0.000 | X | 11.76 | X |
| NDKA_MOUSE | Nme1 | Nucleoside diphosphate kinase A | 5.185 | 0.020 | 1.058 | 0.169 | 29.927 | | 29.41 | |
| NDUS1_MOUSE | Ndufs1 | NADH-ubiquinone oxidoreductase 75 kDa subunit, mitochondrial | 1.859 | 0.005 | 0.967 | 0.283 | 4.623 | | 29.41 | X |
| NFASC_MOUSE | Nfasc | Neurofascin | 1.524 | 0.033 | 1.028 | 0.745 | 0.000 | X | 35.29 | X |
| NRCAM_MOUSE | Nrcam | Neuronal cell adhesion molecule | 1.714 | 0.011 | 1.148 | 0.314 | 0.000 | | 35.29 | X |
| NTRI_MOUSE | Ntm | Neurotrimin | 4.060 | 0.033 | 0.988 | 0.574 | 0.000 | X | 47.06 | X |
| STXB1_MOUSE | Stxbp1 | Syntaxin-binding protein 1 | 1.752 | 0.036 | 1.058 | 0.358 | 0.487 | | 7.056 | X |
| THY1_MOUSE | Thy1 | Thy-1 membrane glycoprotein | 3.357 | 0.022 | 1.069 | 0.541 | 0.487 | | 17.65 | |
| VDAC2_MOUSE | Vdac2 | Voltage-dependent anion-selective channel protein 2 | 2.929 | 0.029 | 0.946 | 0.058 | 14.112 | X | 35.29 | X |

(*Continued*)

**Table 1.** (Continued)

| Protein name (UniProt ID) | Gene name | Description | Palmitome ratio | Palmitome p-value | Proteome ratio | Proteome p-value | CRAPome (%) | Kang and colleagues | Swiss Palm (%) | CSS Palm |
|---|---|---|---|---|---|---|---|---|---|---|
| VISL1_MOUSE | *Vsnl1* | Visinin-like protein 1 | 2.451 | 0.040 | 0.912 | 0.039 | 0.243 | X | 41.18 | X |

X = validation of given dataset; blank cells or a score of zero indicate lack of identification in that dataset [12,31,39,97]. SwissPalm score indicates percentage of palmitoylation experiments in which the protein has been identified.

palmitoylated, although GluA1 has known intracellular and transmembrane palmitoylation sites (C811 and C585, respectively) [51]. Since aberrant palmitoylation of known sites has functional consequences [52,53], we examined whether a GluA1 Cysteine-Alanine (C323A) point mutation, preventing disulfide bonding/palmitoylation at this residue, alters AMPAR function. When expressed in *Xenopus laevis* ooctyes, both with auxiliary subunit TARPγ8 and independently at high concentration, GluA1 C323A exhibits abolished glutamate-evoked AMPAR currents (**Fig 3E**). Intriguingly, GluA2 and AMPAR interactors (AP2A1, AP2B1, AP2M1, NSF, and KCC2D) were also identified as PPT1 substrates (**Fig 3C**).

We used Ingenuity Pathway Analysis (IPA) to determine functional pathways in which high- and medium-confidence PPT1 substrates participate in an unbiased manner. The top pathways identified were (1) GABA receptor signaling; (2) CREB signaling in neurons; (3) synaptic long-term depression; (4) endocannabinoid neuronal synapse pathway; and (5) synaptogenesis signaling pathway (**Fig 3F**). These functional pathways are consistent with *CLN1* phenotypes, and the 9 ontological classes we defined (**Fig 3C**).

## Structural motifs contribute to PPT1 substrate specificity

Since we find that PPT1 is moderately specific and confirmed that it can depalmitoylate 138 proteins *in vivo* (approximately 10% of the synaptic palmitome), we carried out two independent analyses to identify determinants of its substrate specificity. First, we sought to determine whether classes of PPT1 substrates with several high-confidence hits have distinct recognition sites. We discovered that the synaptic adhesion PPT1 substrates are conserved and belong to the Immunoglobulin (Ig) domain-containing class of homotypic adhesion molecules (**Fig 4A**). The best characterized of these are CADM2 (SynCAM2) and NRCAM. Based on their structures, we identified that the carbamidomethyl peptides in most of these proteins map to 1 of 2 crucial cysteines in extracellular IgG domains (**Fig 4B and 4C**) [54]. The palmitoylated cysteines are on exposed and accessible beta-strands and normally form intradomain disulfide bonds to maintain the typical IgG domain fold (**Fig 4C**). The palmitoylated cysteines identified in the other synaptic adhesion molecules are also known to mediate intra- or intermolecular interactions via disulfide bonds [55]. Secreted endogenous PPT1 enzyme activity was detected in the media of WT primary neuronal cultures (**Fig 4D**), suggesting that PPT1 could act in the extracellular space [56]. Decreased colocalization of CADM2 and SYPH in PPT1 KO neurons (**Fig 4E and 4F**) suggests that altered depalmitoylation of synaptic adhesion molecules impacts subsynaptic localization. Overall, our data suggest that PPT1 can recognize the IgG sequence or fold.

Next, we tested whether motifs recognized by partner palmitoylating enzymes are also the primary determinant for PPT1 substrate recognition. Two DHHC/PATs, DHHC5 and DHHC17, localize to synapses [57,58]. DHHC17 and its *Drosophila* homolog, HIP-14, are known to traffic many synaptic proteins in a palmitoylation-dependent manner, including the PPT1 substrate CSPα [8]. DHHC17 is the only PAT with a known substrate-recognition motif

**Table 2. Medium-confidence validated PPT1 substrates.**

| Protein name (UniProt ID) | Gene name | Description | Palmitome ratio | Palmitome p-value | Proteome ratio | Proteome p-value | CRAPome (%) | Kang and colleagues | Swiss Palm (%) | CSS Palm |
|---|---|---|---|---|---|---|---|---|---|---|
| 1433B_MOUSE | Ywhab | 14-3-3 protein beta/alpha | 6.1651 | 0.0345 | 0.9858 | 0.5572 | | | 0 | X |
| ACON_MOUSE | Aco2 | Aconitate hydratase, mitochondrial | 2.376 | 0.027 | 1.026 | 0.810 | 6.569 | | 0 | X |
| ACSF2_MOUSE | Acsf2 | Medium-chain acyl-CoA ligase ACSF2, mitochondrial | 6.8591 | 0.0431 | 1.1975 | 0.0262 | 0.4866 | | 0 | X |
| ACTN1_MOUSE | Actn1 | Alpha-actinin-1 | 2.9415 | 0.0091 | 1.0425 | 0.5896 | | | 0 | |
| ADDG_MOUSE | Add3 | Gamma-adducin | 2.0263 | 0.0064 | 0.9829 | 0.5518 | 7.2993 | | 0 | |
| AIFM1_MOUSE | Aifm1 | Apoptosis-inducing factor 1, mitochondrial | 2.8165 | 0.0430 | 0.9702 | 0.4086 | | | 0 | X |
| AL1L1_MOUSE | Aldh1l1 | Cytosolic 10-formyltetrahydrofolate dehydrogenase | 3.1219 | 0.0122 | 1.0338 | 0.5529 | 0.9732 | | 0 | X |
| ANK3_MOUSE | Ank3 | Ankyrin-3 | 2.4417 | 0.0253 | 1.0168 | 0.8537 | 3.6496 | | 0.487 | |
| AP2A1_MOUSE | Ap2a1 | AP-2 complex subunit alpha-1 | 1.6594 | 0.0299 | 0.9960 | 0.7766 | 5.8394 | | 1.217 | X |
| AP2B1_MOUSE | Ap2b1 | AP-2 complex subunit beta | 2.3904 | 0.0227 | 0.9999 | 0.8375 | | | 2.433 | |
| AP2M1_MOUSE | Ap2m1 | AP-2 complex subunit mu | 2.5018 | 0.0466 | 0.9800 | 0.5819 | 5.1095 | X | 2.433 | X |
| ASTN1_MOUSE | Astn1 | Astrotactin-1 | 1.6647 | 0.0160 | 0.9877 | 0.7745 | 0.2433 | X | 5.353 | X |
| AT2B4_MOUSE | Atp2b4 | Plasma membrane calcium-transporting ATPase 4 | 1.7388 | 0.0114 | 0.9954 | 0.7361 | 3.1630 | | 5.882 | |
| ATAD3_MOUSE | Atad3 | ATPase family AAA domain-containing protein 3 | 1.7738 | 0.0372 | 0.9826 | 0.4550 | | | 5.882 | X |
| ATPG_MOUSE | Atp5f1c | ATP synthase subunit gamma, mitochondrial | 3.3911 | 0.0316 | 1.0012 | 0.7784 | | | 5.882 | X |
| BAIP2_MOUSE | Baiap2 | Brain-specific angiogenesis inhibitor 1-associated protein 2 | 2.5954 | 0.0460 | 0.9172 | 0.0518 | 0.9732 | | 5.882 | |
| CA2D1_MOUSE | Cacna2d1 | Voltage-dependent calcium channel subunit alpha-2/delta-1 | 3.1679 | 0.0182 | 0.9515 | 0.2687 | 0.7299 | | 11.76 | X |
| CAPS1_MOUSE | Cadps | Calcium-dependent secretion activator 1 | 2.4798 | 0.0483 | 0.9837 | 0.5833 | 0.4866 | | 17.65 | X |
| CBPE_MOUSE | Cpe | Carboxypeptidase E | 3.1240 | 0.0467 | 1.0509 | 0.2354 | 0.0000 | | 17.65 | X |
| CCG8_MOUSE | Cacng8 | Voltage-dependent calcium channel gamma-8 subunit | 3.1719 | 0.0204 | 0.9912 | 0.6364 | 0.0000 | X | 22.14 | X |
| CD47_MOUSE | Cd47 | Leukocyte surface antigen CD47 | 2.9829 | 0.0084 | 0.9688 | 0.3699 | 0.0000 | X | 29.41 | |
| CISY_MOUSE | Cs | Citrate synthase, mitochondrial | 2.9715 | 0.0258 | 1.0134 | 0.8591 | | | 47.06 | |
| CLH1_MOUSE | Cltc | Clathrin heavy chain 1 | 2.8044 | 0.0232 | 0.9978 | 0.8026 | | | 52.94 | |
| CLUS_MOUSE | Clu | Clusterin | 7.8734 | 0.0278 | 1.0804 | 0.3769 | 1.7032 | | 58.82 | X |
| COTL1_MOUSE | Cotl1 | Coactosin-like protein | 2.2434 | 0.0267 | 1.0294 | 0.0758 | 0.2433 | X | 0 | X |
| COX2_MOUSE | Mtco2 | Cytochrome c oxidase subunit 2 | 1.6676 | 0.0472 | 1.0305 | 0.5074 | | | 0 | |
| COX41_MOUSE | Cox4i1 | Cytochrome c oxidase subunit 4 isoform 1, mitochondrial | 8.0469 | 0.0285 | 0.9785 | 0.4664 | 4.3796 | X | 0 | |
| CRIP2_MOUSE | Crip2 | Cysteine-rich protein 2 | 6.8277 | 0.0496 | 0.9438 | 0.1959 | 0.4866 | | 0 | X |
| CTNA2_MOUSE | Ctnna2 | Catenin alpha-2 | 1.8749 | 0.0463 | 1.0109 | 0.8808 | 3.1630 | | 0 | X |
| CYFP1_MOUSE | Cyfip1 | Cytoplasmic FMR1-interacting protein 1 | 11.9841 | 0.0440 | 1.0567 | 0.3692 | | | 0 | X |
| DCE1_MOUSE | Gad1 | Glutamate decarboxylase 1 | 2.8429 | 0.0430 | 1.0195 | 0.8945 | 0.0000 | | 0 | |
| DCE2_MOUSE | Gad2 | Glutamate decarboxylase 2 | 1.877 | 0.050 | 1.052 | 0.246 | 0.000 | | 0 | X |
| DCLK1_MOUSE | Dclk1 | Serine/threonine-protein kinase DCLK1 | 2.6248 | 0.0212 | 1.0263 | 0.7258 | 0.4866 | | 0 | X |

*(Continued)*

**Table 2.** (Continued)

| Protein name (UniProt ID) | Gene name | Description | Palmitome ratio | Palmitome p-value | Proteome ratio | Proteome p-value | CRAPome (%) | Kang and colleagues | Swiss Palm (%) | CSS Palm |
|---|---|---|---|---|---|---|---|---|---|---|
| DCTN1_MOUSE | Dctn1 | Dynactin subunit 1 | 1.5601 | 0.0173 | 0.9741 | 0.3428 | | | 0 | X |
| DEST_MOUSE | Dstn | Destrin | 4.3473 | 0.0220 | 0.9684 | 0.4662 | | | 0 | X |
| DMXL2_MOUSE | Dmxl2 | DmX-like protein 2 | 1.6352 | 0.0331 | 0.9837 | 0.5503 | 0.2433 | | 0 | X |
| DNJC5_MOUSE | Dnajc5 | DnaJ homolog subfamily C member 5 | 5.1200 | 0.0109 | 0.8396 | 0.0177 | 0.0000 | X | 0 | X |
| DPP6_MOUSE | Dpp6 | Dipeptidyl aminopeptidase-like protein 6 | 1.5302 | 0.0185 | 0.9976 | 0.7721 | 0.2433 | | 0 | |
| DYN1_MOUSE | Dnm1 | Dynamin-1 | 2.3159 | 0.0428 | 0.9753 | 0.4249 | 4.6229 | | 0.243 | X |
| EFTU_MOUSE | Tufm | Elongation factor Tu, mitochondrial | 1.5273 | 0.0058 | 0.9989 | 0.7913 | | | 0.487 | X |
| EP15R_MOUSE | Eps15l1 | Epidermal growth factor receptor substrate 15-like 1 | 1.8051 | 0.0421 | 0.9319 | 0.1354 | 6.5693 | | 0.487 | |
| ERP29_MOUSE | Erp29 | Endoplasmic reticulum resident protein 29 | 17.3706 | 0.0265 | 1.0100 | 0.9527 | 6.5693 | | 0.73 | X |
| F10A1_MOUSE | St13 | Hsc70-interacting protein | 2.0988 | 0.0335 | 0.9517 | 0.2199 | | | 0.973 | X |
| FA49B_MOUSE | Fam49b | Protein FAM49B | 2.0721 | 0.0411 | 0.9483 | 0.2902 | 0.9732 | | 0.973 | |
| FAS_MOUSE | Fasn | Fatty acid synthase | 2.0774 | 0.0122 | 1.0023 | 0.9295 | | | 1.217 | X |
| FUMH_MOUSE | Fh | Fumarate hydratase, mitochondrial | 2.4212 | 0.0422 | 1.0295 | 0.6925 | 5.8394 | | 2.433 | |
| GNAI1_MOUSE | Gnai1 | Guanine nucleotide-binding protein G(i) subunit alpha-1 | 3.3180 | 0.0287 | 0.9355 | 0.2010 | | X | 3.406 | X |
| GNAI2_MOUSE | Gnai2 | Guanine nucleotide-binding protein G(i) subunit alpha-2 | 4.6764 | 0.0374 | 0.9933 | 0.6470 | | X | 3.406 | X |
| GNAO_MOUSE | Gnao1 | Guanine nucleotide-binding protein G(o) subunit alpha | 4.8055 | 0.0254 | 1.0334 | 0.7333 | | X | 3.893 | X |
| GNAQ_MOUSE | Gnaq | Guanine nucleotide-binding protein G(q) subunit alpha | 3.9366 | 0.0329 | 1.0152 | 0.9241 | 2.9197 | X | 4.136 | |
| GNAZ_MOUSE | Gnaz | Guanine nucleotide-binding protein G(z) subunit alpha | 2.8957 | 0.0265 | 0.9815 | 0.5618 | 1.4599 | X | 4.38 | X |
| GPDA_MOUSE | Gpd1 | Glycerol-3-phosphate dehydrogenase | 2.0714 | 0.0477 | 0.9542 | 0.2820 | 0.0000 | | 5.109 | X |
| GRIA2_MOUSE | Gria2 | Glutamate receptor 2 (GluR-2) | 2.0033 | 0.0397 | 0.9980 | 0.8023 | 0.0000 | X | 5.882 | |
| HEXB_MOUSE | Hexb | Beta-hexosaminidase subunit beta | 2.2503 | 0.0402 | 1.2536 | 0.0001 | 1.7032 | | 5.882 | |
| HXK1_MOUSE | Hk1 | Hexokinase-1 | 1.864 | 0.032 | 1.041 | 0.483 | 9.002 | | 8.516 | X |
| HS12A_MOUSE | Hspa12a | Heat shock 70 kDa protein 12A | 2.0260 | 0.0458 | 1.0011 | 0.8019 | 0.2433 | | 5.882 | X |
| HSP74_MOUSE | Hspa4 | Heat shock 70 kDa protein 4 | 2.0908 | 0.0249 | 0.9931 | 0.7273 | | | 7.299 | X |
| IMPA1_MOUSE | Impa1 | Inositol monophosphatase 1 | 3.8661 | 0.0392 | 0.9405 | 0.0634 | 2.1898 | X | 8.516 | X |
| IVD_MOUSE | Ivd | Isovaleryl-CoA dehydrogenase, mitochondrial | 1.6243 | 0.0289 | 1.0186 | 0.5851 | 2.4331 | | 11.76 | X |
| KCC2D_MOUSE | Camk2d | Calcium/calmodulin-dependent protein kinase type II subunit delta | 1.5646 | 0.0170 | 0.9913 | 0.7451 | 3.6496 | | 11.76 | X |
| KIF5C_MOUSE | Kif5c | Kinesin heavy chain isoform 5C | 1.6691 | 0.0351 | 0.9917 | 0.6342 | | | 11.76 | X |
| LETM1_MOUSE | Letm1 | Mitochondrial proton/calcium exchanger protein | 1.8560 | 0.0428 | 0.9614 | 0.2821 | 4.3796 | | 11.76 | X |
| LIPA3_MOUSE | Ppfia3 | Liprin-alpha-3 | 1.6859 | 0.0239 | 0.9300 | 0.1882 | 0.2433 | | 11.76 | X |

(*Continued*)

**Table 2.** (Continued)

| Protein name (UniProt ID) | Gene name | Description | Palmitome ratio | Palmitome *p*-value | Proteome ratio | Proteome *p*-value | CRAPome (%) | Kang and colleagues | Swiss Palm (%) | CSS Palm |
|---|---|---|---|---|---|---|---|---|---|---|
| LIS1_MOUSE | *Pafah1b1* | Platelet-activating factor acetylhydrolase IB subunit alpha | 2.0802 | 0.0263 | 0.9655 | 0.3596 | 4.3796 | | 16.55 | |
| M2OM_MOUSE | *Slc25a11* | Mitochondrial 2-oxoglutarate/malate carrier | 3.1066 | 0.0359 | 1.0222 | 0.8429 | | | 17.65 | |
| MPP2_MOUSE | *Mpp2* | MAGUK p55 subfamily member 2 | 2.3788 | 0.0249 | 1.1196 | 0.0430 | 0.0000 | X | 17.65 | |
| NAC1_MOUSE | *Slc8a1* | Sodium/calcium exchanger 1 | 1.6078 | 0.0206 | 1.0694 | 0.2621 | 0.2433 | X | 23.53 | X |
| NCALD_MOUSE | *Ncald* | Neurocalcin-delta | 45.0986 | 0.0333 | 0.8888 | 0.0594 | 0.0000 | | 23.53 | X |
| NCKP1_MOUSE | *Nckap1* | Nck-associated protein 1 | 1.7610 | 0.0369 | 1.0207 | 0.6890 | 8.5158 | | 25.06 | |
| NRX3A_MOUSE | *Nrxn3* | Neurexin-3 | 2.8629 | 0.0254 | 1.0057 | 0.9531 | 0.0000 | | 41.18 | X |
| NSF_MOUSE | *Nsf* | Vesicle-fusing ATPase | 1.7968 | 0.0111 | 0.9932 | 0.6500 | 7.5426 | | 41.18 | X |
| OAT_MOUSE | *Oat* | Ornithine aminotransferase, mitochondrial | 2.3834 | 0.0408 | 1.1231 | 0.2428 | | | 47.06 | |
| ODO1_MOUSE | *Ogdh* | 2-oxoglutarate dehydrogenase, mitochondrial | 2.7338 | 0.0474 | 1.0213 | 0.8685 | 7.7859 | | 47.06 | X |
| ODPX_MOUSE | *Pdhx* | Pyruvate dehydrogenase protein X component, mitochondrial | 1.5552 | 0.0191 | 0.9543 | 0.0361 | 2.9197 | | 52.94 | |
| OPA1_MOUSE | *Opa1* | Dynamin-like 120 kDa protein, mitochondrial | 1.9052 | 0.0425 | 0.9759 | 0.3227 | 3.1630 | X | 64.71 | X |
| OXR1_MOUSE | *Oxr1* | Oxidation resistance protein 1 | 1.6551 | 0.0027 | 0.9682 | 0.3713 | 0.0000 | | 0 | |
| PCCA_MOUSE | *Pcca* | Propionyl-CoA carboxylase alpha chain, mitochondrial | 1.5242 | 0.0322 | 0.9542 | 0.2035 | | | 0 | X |
| PDE2A_MOUSE | *Pde2a* | cGMP-dependent 3',5'-cyclic phosphodiesterase | 1.7097 | 0.0100 | 1.0481 | 0.3392 | 0.7299 | X | 0.243 | X |
| PGM1_MOUSE | *Pgm1* | Phosphoglucomutase-1 | 1.9775 | 0.0013 | 1.0370 | 0.6444 | 3.8929 | | 2.433 | |
| PLCB1_MOUSE | *Plcb1* | 1-phosphatidylinositol 4,5-bisphosphate phosphodiesterase beta-1 | 1.9949 | 0.0198 | 1.0061 | 0.7702 | 0.0000 | | 5.882 | X |
| PP2BA_MOUSE | *Ppp3ca* | Serine/threonine-protein phosphatase 2B catalytic subunit alpha isoform | 1.7693 | 0.0242 | 0.9565 | 0.1996 | 1.7032 | X | 11.76 | |
| PRDX6_MOUSE | *Prdx6* | Peroxiredoxin-6 | 1.6130 | 0.0299 | 1.0689 | 0.0748 | | X | 17.65 | |
| PRRT3_MOUSE | *Prrt3* | Proline-rich transmembrane protein 3 | 2.2877 | 0.0290 | 0.9837 | 0.5961 | 0.2433 | | 17.65 | X |
| PTPRD_MOUSE | *Ptprd* | Receptor-type tyrosine-protein phosphatase delta | 2.8642 | 0.0307 | 1.0097 | 0.9828 | 0.0000 | X | 17.65 | X |
| PTPRS_MOUSE | *Ptprs* | Receptor-type tyrosine-protein phosphatase S | 2.2591 | 0.0261 | 0.9756 | 0.4610 | 0.0000 | | 29.41 | |
| PYGB_MOUSE | *Pygb* | Glycogen phosphorylase, brain form | 3.9945 | 0.0321 | 1.0839 | 0.0150 | 4.6229 | | 35.29 | |
| RAB1A_MOUSE | *Rab1a* | Ras-related protein Rab-1A | 3.9014 | 0.0486 | 0.9073 | 0.0628 | | | 37.96 | X |
| RAB7A_MOUSE | *Rab7a* | Ras-related protein Rab-7a | 1.8488 | 0.0354 | 1.0068 | 0.8316 | | X | 41.18 | X |
| RAP2B_MOUSE | *Rap2b* | Ras-related protein Rap-2b | 8.6218 | 0.0214 | 0.9733 | 0.4554 | 1.7032 | X | 0 | X |
| S12A2_MOUSE | *Slc12a2* | Solute carrier family 12 member 2 | 2.5952 | 0.0396 | 0.9641 | 0.3628 | 1.2165 | | 0 | X |
| SAHH2_MOUSE | *Ahcyl1* | S-adenosylhomocysteine hydrolase-like protein 1 | 2.7124 | 0.0136 | 1.0495 | 0.2339 | 3.8929 | | 0 | X |
| SATT_MOUSE | *Slc1a4* | Neutral amino acid transporter A | 8.4285 | 0.0440 | 1.2718 | 0.0002 | 0.0000 | X | 0.243 | X |

(*Continued*)

**Table 2.** (Continued)

| Protein name (UniProt ID) | Gene name | Description | Palmitome ratio | Palmitome *p*-value | Proteome ratio | Proteome *p*-value | CRAPome (%) | Kang and colleagues | Swiss Palm (%) | CSS Palm |
|---|---|---|---|---|---|---|---|---|---|---|
| SC6A1_MOUSE | Slc6a1 | Sodium- and chloride-dependent GABA transporter 1 | 3.4145 | 0.0466 | 1.0470 | 0.4440 | 0.0000 | X | 0.973 | X |
| SCAM1_MOUSE | Scamp1 | Secretory carrier-associated membrane protein 1 | 3.2759 | 0.0336 | 0.9115 | 0.0531 | 0.7299 | X | 1.46 | X |
| SDHB_MOUSE | Sdhb | Succinate dehydrogenase [ubiquinone] iron-sulfur subunit, mitochondrial | 2.8442 | 0.0205 | 1.0124 | 0.8729 | 4.3796 | | 2.676 | X |
| SEPT8_MOUSE | Septin8 | Septin-8 | 1.7848 | 0.0199 | 1.0574 | 0.3595 | | X | 5.109 | |
| SFXN1_MOUSE | Sfxn1 | Sideroflexin-1 | 7.9333 | 0.0425 | 1.0657 | 0.3687 | 6.5693 | X | 5.882 | X |
| SFXN3_MOUSE | Sfxn3 | Sideroflexin-3 | 2.3725 | 0.0162 | 0.9884 | 0.6007 | 0.9732 | X | 5.882 | X |
| SSDH_MOUSE | Aldh5a1 | Succinate-semialdehyde dehydrogenase, mitochondrial | 1.6663 | 0.0050 | 1.0534 | 0.4846 | 4.3796 | | 5.882 | X |
| SYIM_MOUSE | Iars2 | Isoleucine—tRNA ligase, mitochondrial | 1.8534 | 0.0348 | 1.0386 | 0.5893 | 4.3796 | | 8.516 | X |
| SYNJ1_MOUSE | Synj1 | Synaptojanin-1 | 2.1789 | 0.0289 | 1.0211 | 0.8514 | 2.9197 | | 9.489 | X |
| SYNPR_MOUSE | Synpr | Synaptoporin | 6.445 | 0.018 | 0.963 | 0.365 | 0.000 | | 11.76 | X |
| SYT2_MOUSE | Syt2 | Synaptotagmin-2 | 2.4842 | 0.0494 | 0.9862 | 0.7697 | 0.0000 | X | 11.76 | X |
| SYT7_MOUSE | Syt7 | Synaptotagmin-7 | 2.7289 | 0.0129 | 1.0495 | 0.6112 | 0.0000 | X | 11.76 | X |
| TERA_MOUSE | Vcp | Transitional endoplasmic reticulum ATPase) | 2.1826 | 0.0257 | 1.0105 | 0.9593 | 0.4063 | | 14.36 | |
| UBA1_MOUSE | Uba1 | Ubiquitin-like modifier-activating enzyme 1 | 2.3035 | 0.0188 | 0.9517 | 0.2816 | 0.2603 | | 23.53 | X |
| UCRI_MOUSE | Uqcrfs1 | Cytochrome b-c1 complex subunit Rieske, mitochondrial | 4.8396 | 0.0492 | 1.1149 | 0.0848 | 2.4331 | X | 28.71 | |
| VA0D1_MOUSE | Atp6v0d1 | V-type proton ATPase subunit d | 4.6384 | 0.0334 | 1.0246 | 0.7738 | 1.2165 | | 29.41 | X |
| VAMP1_MOUSE | Vamp1 | Vesicle-associated membrane protein 1 | 6.0833 | 0.0172 | 0.8937 | 0.0789 | 0.0000 | X | 29.41 | |
| VATE1_MOUSE | Atp6v1e1 | V-type proton ATPase subunit E | 2.8166 | 0.0390 | 0.9533 | 0.1839 | 3.8929 | | 29.41 | |
| VPS35_MOUSE | Vps35 | Vacuolar protein sorting-associated protein 35 | 2.2835 | 0.0102 | 0.9934 | 0.6715 | 8.5158 | | 47.06 | |
| VTA1_MOUSE | Vta1 | Vacuolar protein sorting-associated protein VTA1 homolog | 39.3736 | 0.0420 | 0.9281 | 0.0333 | 2.1898 | | 0 | |

X = validation of given dataset; blank cells or a score of zero indicate lack of identification in that dataset [12,31,39,97]. SwissPalm score indicates percentage of palmitoylation experiments in which the protein has been identified.

[2]. Its recognition sequence ([VIAP][VIT]xxQP) is usually adjacent to the palmitoylated cysteine. We screened the 138 validated PPT1 substrates for those containing the DHHC17 recognition motif and identified 9 cytosolic proteins, including CLH1 (Clathrin heavy chain), DYN1 (Dynamin 1), SYNJ1 (Synaptojanin 1), and DNJC5 (CSPα) (**Table 2**; **S3A Fig**). IPA identified "Clathrin-mediated endocytosis signaling" as the top pathway involving this subset of PPT1 substrates (**S3B Fig**), consistent with the known function these proteins play in synaptic vesicle endocytosis [59] and alterations in endocytosis observed with *Ppt1* mutations in *Drosophila* [60]. Together, these data suggest that the DHHC17 recognition site is a determinant of substrate specificity for the endocytic class of PPT1 substrates.

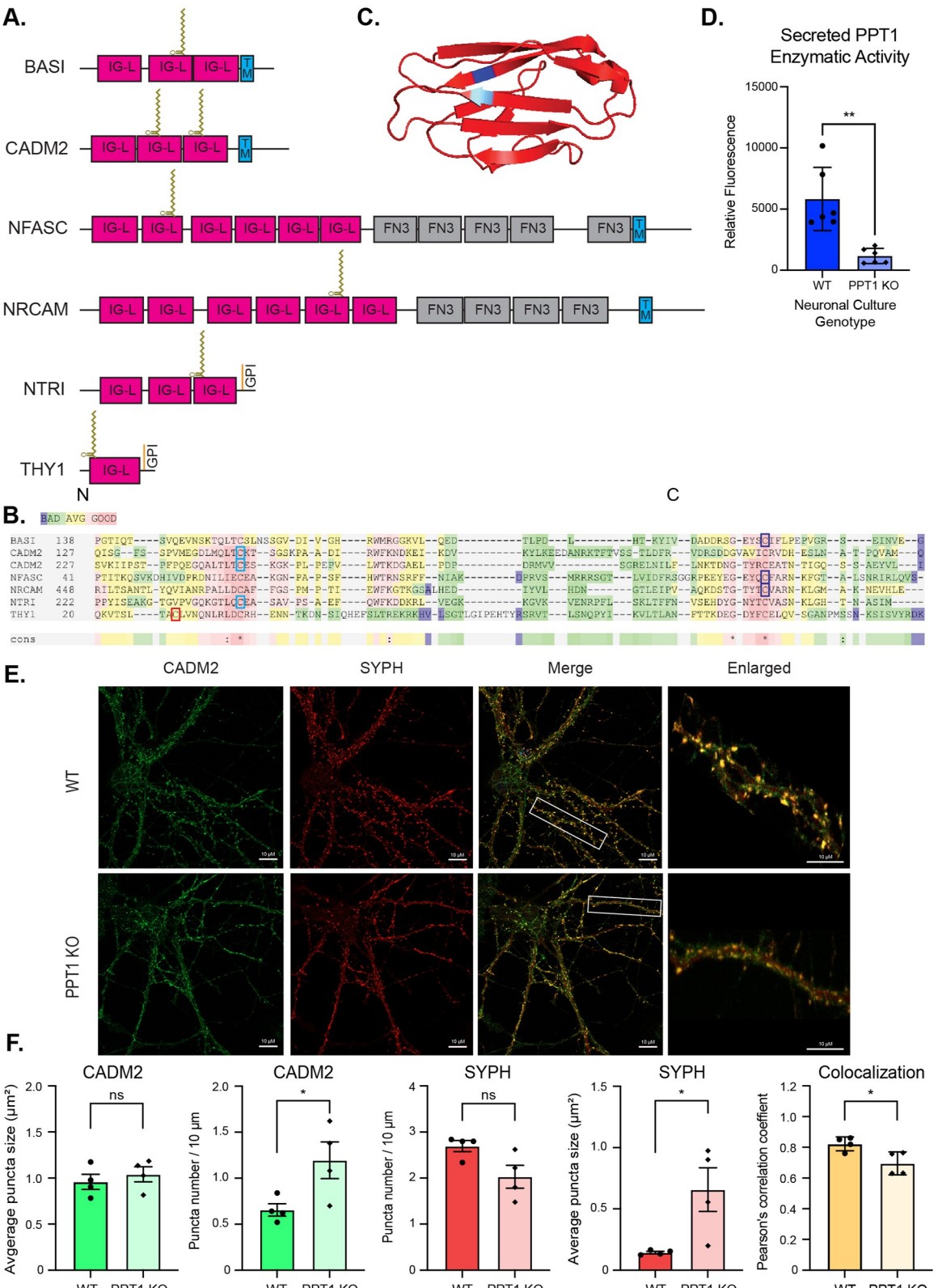

**Fig 4. PPT1 depalmitoylates the IgG domain of synaptic adhesion molecules. 4. (A)** Domain organization of IgG domain-containing synaptic adhesion molecules identified as high-confidence PPT1 substrates with palmitoylation sites represented as lipid chains. IG-L–Ig-like; FN3 –fibronectin 3; TM–transmembrane. **(B)** IgG domain-containing synaptic adhesion molecules show high homology

surrounding the palmitoylated cysteines. Red indicates high homology; green indicates low homology. The 2 conserved cysteines at the N-terminal (light blue) and carboxyl terminus (dark blue) of this domain are both identified as carbamidomethylated in different experiments. **(C)** Representative IgG domain structure (CADM2; PDB, 3M45) with position of palmitoylated cysteines highlighted in blue. **(D)** Activity of PPT1 enzyme measured from WT or PPT1 KO primary neuronal culture medium. Data presented as mean relative fluorescence normalized to total protein concentration ± SD ($n$ = 6 cultures per genotype; ** $p$ > 0.01; **S1 Data**). **(E)** Endogenous syncam2 (CADM2) and synaptophysin 1 (SYPH) colocalize on MAP2+ neurites in WT and PPT1 KO mouse primary neuronal cultures (Scale bars: 10 μm; enlarged ROIs). **(F)** Quantifications of syncam2 and synaptophysin 1 puncta number (per 10-μm neurite segment), average puncta size (μm$^2$), and colocalization (Pearson correlation coefficient). Data presented as mean of 5 ROIs per culture ± SD ($n$ = 4 cultures; * $p$ > 0.5; **S1 Data**). Ig, Immunoglobulin; KO, knockout; PPT1, palmitoyl protein thioesterase 1; ROS, region of interest; WT, wild-type.

## Tertiary PTM screen detects coincidence of palmitoylation and disulfide bonds

To more directly query the relationship between disulfide bond formation and palmitoylation among validated PPT1 substrates, we performed a modified Acyl RAC screen of synaptosomes prepared from 3 WT and 3 PPT1 KO mouse brains (each in technical triplicate; age = 2 months; **Fig 5A**). First, free thiols were labeled with NEM (**Fig 5Ai**), then TCEP was used to reduce disulfide bonds to free thiols (**Fig 5Aii**). The newly generated thiol groups were then labeled with d₅-*N*-ethylmaleimide (dNEM), which has +5 molecular weight shift due to deuterium [61], allowing for the differentiation of cysteines that existed as free thiols versus those in disulfide bonds (**Fig 5Aii**). The remainder of the Acyl RAC workflow was then conducted as previously described (**Fig 1B**) to isolate the synaptic palmitome for LFQ-MS/MS. Carbamidomethyl peptides were detected for 100% (26/26) of high-confidence substrates (**S5 Table**), 59% (66/112) of medium-confidence substrates, and 0% of residual substrates in this tertiary screen, indicating excellent reproducibility and highlighting the efficacy of our stringent confidence thresholds.

To examine PTMs, we aggregated peptide data from the 3 biological replicates to capture as many modified peptides per protein as possible ($n$ = 16,381 total peptides; $n$ = 1,419 unique proteins). We then filtered peptides by individual PTMs of interest (**Fig 5B**): carbamidomethyl (carba.) indicates the presence of a palmitate group, NEM indicates a free thiol, dNEM indicates a disulfide bond, and Other/None describes unmodified peptides or those with a PTM not pertinent to this experiment (**Fig 5C**). Carbamidomethyl modifications were the most abundant PTM of interest (WT = 19.2 ± 0.9%; PPT1 KO = 23.0 ± 1.3%). As expected, carbamidomethyl peptides were significantly more abundant in the PPT1 KO ($p$ = 0.0147), reflecting the increase in palmitoylated substrates due to PPT1 deficiency. Other/None peptides exhibited a concomitant significant decrease in the PPT1 KO (WT = 74.6 ± 0.83%; PPT1 KO = 70.60 ± 1.44%; $p$ = 0.0142). NEM peptides were also abundant but did not display significant changes between genotypes (WT = 6.2 ± 0.2%; PPT1 KO = 6.3 ± 0.1%; $p$ = 0.4688). dNEM-modified peptides were rarest, constituting less than 1% of the total normalized peptide abundance (WT = 0.83 ± 0.03%; PPT1 KO = 0.97 ± 0.1%; $p$ = 0.1302; **Fig 5C**).

We next examined instances where multiple PTMs are found to modify the same cysteine residue (**Fig 5D–5F**) by comparing the lists of peptides curated by modification. We identified sequences present in multiple lists (**Fig 5B**), then examined the ratio of the coincident PTMs (**Fig 5D–5F**). A significant increase in carba.:dNEM was observed in PPT1 KO (WT = 8.02 ± 0.53; PPT1 KO = 9.55 ± 0.25; $p$ = 0.0108; **Fig 5D**), indicating the persistence of palmitoyl groups on cysteines that form disulfide bonds. We did not observe significant changes to carba.:NEM in PPT1 KO (WT = 3.10 ± 0.39; PPT1 KO = 3.50 ± 0.11; $p$ = 0.1669; **Fig 5E**), suggesting that excess palmitoylation alone does not significantly affect the proportion of free thiols. dNEM:NEM decreased significantly in PPT1 KO (WT = 0.83 ± 0.04; PPT1 KO = 0.61 ± 0.12%, $p$ = 0.0385; **Fig 5F**). Together, these data suggest that there is a persistence of palmitates in PPT1 KO on disulfide bonding cysteines.

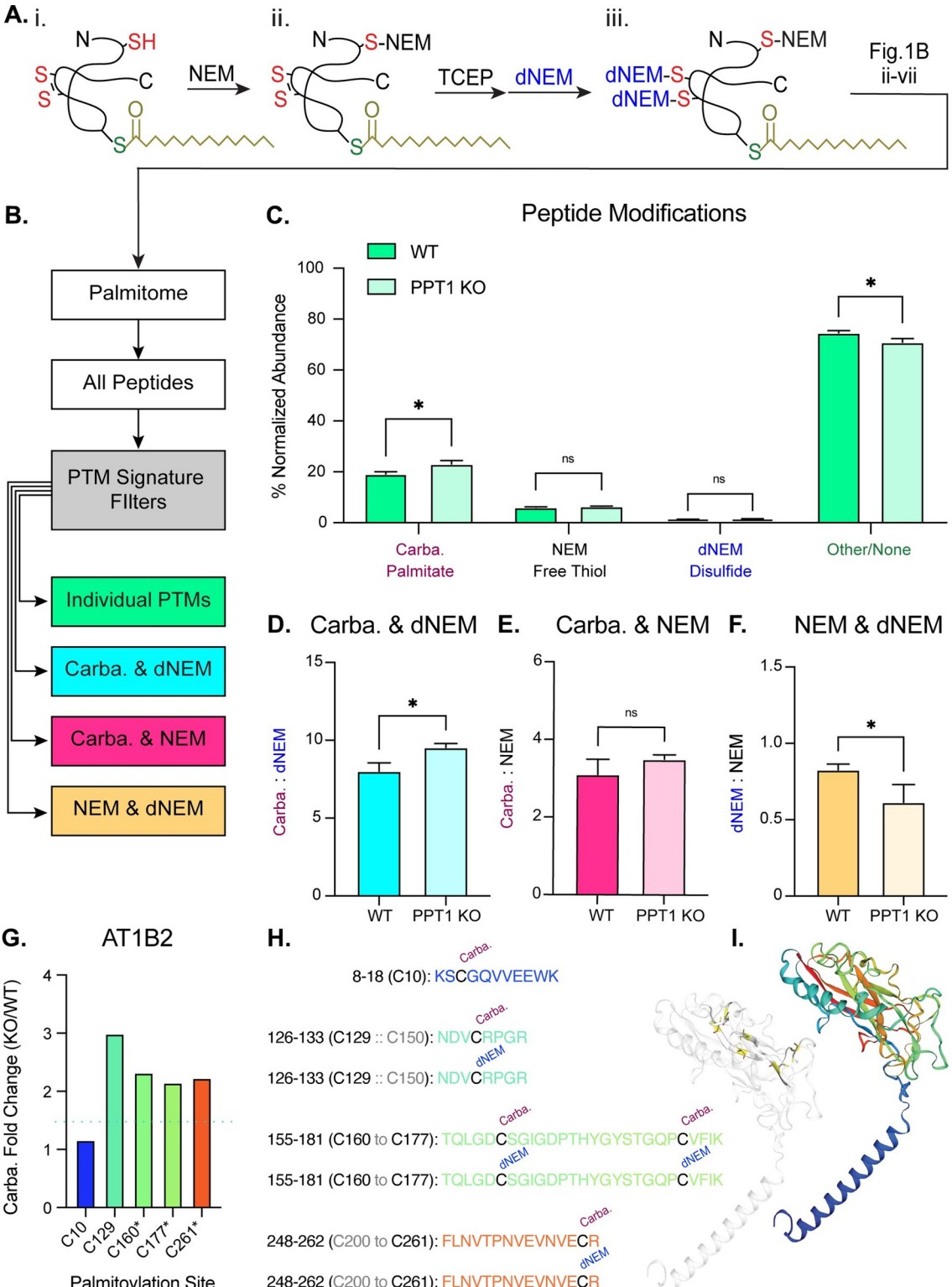

**Fig 5. PPT1-mediated depalmitoylation facilitates disulfide bond formation. (A)** Schematic of modified Acyl RAC. (i) Free thiols were blocked with NEM prior to reduction of disulfide bonds with TCEP. (ii) Free thiols generated by TCEP treatment were modified with heavy-labeled NEM (dNEM). iii. Acyl RAC was then conducted as previously described (**Fig 1Bii–iv**). **(B)** Pipeline for PTM analysis of Acyl RAC peptide data. Peptides were aggregated from 3 independent proteomic screens with 3 biological replicates. All peptides were filtered by individual PTMs or by coincidence signatures where 2 distinct PTMs were found to modify the same cysteine

residue. **(C)** Global distribution of modified peptides. Bars represent percent (%) average normalized abundance of the different peptide classes. The value of each replicate is calculated as the summed abundance of peptides with the given PTM normalized to the total abundance for all detected peptides (**S1 Data**). Cysteine residues with PTMs are interpreted as follows: carbamidomethyl (carba.) = palmitate, NEM = free thiol, dNEM = disulfide bond, other or no moieties (Other/None). **(D–F)** Ratios of PTM normalized abundance for peptides where 2 PTMs are found to modify the same cysteine residue. The value of each replicate is calculated as the summed abundance of peptides with the first PTM divided by the summed normalized abundance of peptides with the second PTM (mean ± SD; * $p < 0.05$, unpaired, 2-tailed $t$ test; **S1 Data**). **(G)** Peptide analysis of a high-confidence PPT1 substrate, Sodium/ Potassium transporting ATPase subunit β1 (AT1B2). Normalized abundance fold change (PPT1 KO/WT) of carbamidomethyl AT1B2 peptides (1.5-fold change indicated by blue dashed line). Palmitoylated cysteine sites marked with an asterisk (*) were confirmed to be PPT1 substrates in the validation screen. There is a persistence of palmitates on cysteines that normally participate in disulfide bonds in PPT1 KO (C129, C160, C177, and C261; **S5 Table**), while C10, which is not predicted to form a disulfide linkage, appears unchanged (**S1 Data**). **(H)** Detected AT1B2 peptides are indicated. For those sequences with cysteines predicted to form disulfide bonds, carba. and dNEM moieties were found to modify that cysteine residue. **(I)** Ribbon structure of AT1B2 (SWISS-MODEL; P14231) with disulfide bonds highlighted. Bar graph and peptide sequences are color coded according to location in the ribbon structure. Acyl RAC, Acyl Resin-Assisted Capture; dNEM, d5-N-ethylmaleimide; KO, knockout; NEM, N-ethylmaleimide; PPT1, palmitoyl protein thioesterase 1; PTM, posttranslational modification; TCEP, tris(2-carboxyethyl)phosphine; WT, wild-type.

We next examined our curated peptide modification data with a focus on high-confidence PPT1 substrates. Carbamidomethyl moieties were detected on 94% (17/18) of validated PPT1 depalmitoylation sites predicted to form a disulfide bond (sole exception is NRCAM C519; **S5 Table**). This result again highlights the reproducibility of our method and corroborates our previous observation that palmitoylation and disulfide bond formation may have a functional relationship. dNEM moieties, validating the presence of a disulfide bond, were found on 33% (6/18) of these sites (AT1B1 C126, AT1B2 C160 and C177, AT1B2 C261, GPM6A C192, and THY1 C28; **S5 Table**). Since 3 of these sites were from a single protein, the Na$^+$/K$^+$ transporting ATPase subunit β2 (AT1B2), we chose to investigate the peptide data for this protein in greater detail (**Fig 5G–5I**). Four unique AT1B2 peptides with 5 carbamidomethylated cysteines (C10, C129, C160, C177, and C261) were identified in the tertiary screen (**Fig 5G and 5H**). C10, which is not predicted to form a disulfide bond, was previously identified as a palmitoylation site [11]. The other palmitoylation sites are novel to the best of our knowledge and are predicted to form disulfide bonds (**S5 Table**). Indeed, each of these sites was found to be modified by dNEM. C160, C177, and C261 are high-confidence PPT1 substrate sites (indicated by an asterisk in **Fig 5G**), while C10 and C129 did not meet this criterion in the initial screens. We examined normalized abundance fold change (PPT1 KO/WT) of AT1B2 carbamidomethyl peptides and observed a persistence of palmitate groups on cysteines that normally participate in disulfide bonds in PPT1 KO (**Fig 5G**). This result corroborates our data that C160, C177, and C261 are PPT1 depalmitoylation sites and suggests that C129 may be as well, while C10 is likely not. Examination of the crystal structure of AT1B2 reveals that C129, C160, C177, and C261 account for all disulfide linkages in this molecule which serve to stabilize β-sheets that form the channel pore (**Fig 5I**) [62].

## Discussion

We systematically and quantitatively characterized palmitoylation in WT and PPT1 KO mice and identified and validated PPT1 substrates. The broad and quantitative nature of our approach also enabled us to illuminate novel functions of depalmitoylation at the synapse. Here, we highlight the most salient insights revealed by our analyses to encourage others to use this resource in their own investigations.

### Generation of brain and synaptic palmitomes

In this study, we present a comprehensive dataset of brain ($n = 1,795$) and synaptic palmitomes ($n = 1,378$). This is the largest mouse brain palmitome to date and is a valuable resource to the scientific community (PRIDE accession PXD032052).

## Stringent validation of PPT1 substrates

As a result of our 2-step screen, we identified 138 high- to medium-confidence PPT1 substrates. We are convinced that these proteins are bona fide substrates due to the stringent nature of our selection process: (1) only proteins that were independently identified in 3 biological samples with robust identification measures were included (>2 unique peptides; confidence score > 100); (2) *in vivo* differences (KO/WT >1.5; $p < 0.05$) were utilized as initial criteria for selection (**Fig 2**, **S1 Fig**); (3) putative substrates were independently validated by direct PPT1 enzymatic depalmitoylation (**Fig 3**); (4) the tertiary screen confirmed the majority of the high- to medium-confidence hits and, importantly, none of the nonvalidated or residual hits (**Fig 5**); (5) previously published PPT1 substrates were validated (**Tables 1 and 2**); (6) palmitoylated proteins known to be exclusively depalmitoylated by AHBD-17 or APT1 were not identified, indicating specificity; (7) the PPT1 substrates cluster into known functional groups (**Fig 3**); and (8) synaptic adhesion molecules and endocytic PPT1 substrates have common structural motifs that explain substrate specificity (**Fig 4**, **S3 Fig**).

Collectively, our data indicate that PPT1 is a moderately selective enzyme that depalmitoylates approximately 10% of the synaptic palmitome, while other depalmitoylating enzymes (APTs, ABHDs) likely depalmitoylate the remaining palmitoylated proteins. As with any proteome-wide screen, a fraction of PPT1 substrates was likely not identified, possibly among residual hits (**S4 Table**) and should be reexamined in a case-by-case manner. Furthermore, PPT1 and other depalmitoylating enzymes may have overlapping substrates, although our results suggest this is uncommon, as the protein expression of APTs and ABHDs does not compensate for loss of PPT1 (**Fig 2D**, **S1D Fig**).

## Elucidation of the relationship between PPT1-mediated depalmitoylation and protein expression

PPT1 depalmitoylation activity has long been considered necessary for protein degradation [63,64], as *CLN1* patient-derived cells exhibit accumulation of lipid modified proteins [64,65]. Having successfully identified substrates of PPT1, we surprisingly find that in PPT1 KO, most substrates show significantly increased palmitoylation independently from increased protein expression, and exhibit either unchanged or decreased protein levels at the synapse (**Fig 2H**). This analysis suggests that for most proteins, removal of the palmitate group by PPT1 is either not required for proteasomal degradation, or another depalmitoylating enzyme facilitates this process at early-stage disease. The noteworthy exceptions are the group of proteins with both increased expression and palmitoylation ($n = 4$; ASAH1, CATD, SCRB2, and TPP1; **Fig 2H**), which precluded us from identifying them as PPT1 substrates in this study. TPP1 and CATD accumulation were previously observed in a mouse brain lysosomal proteome of late-stage *CLN1* [39] and ASAH1 and TPP1 accumulate in both early- and late-stage *CLN1* spinal cord [36], suggesting that these changes are relevant to disease pathogenesis. It is possible that these proteins exhibit depalmitoylation-dependent degradation, as previously suggested to explain lipidated protein accumulation in *CLN1* disease [65]. The contributions of ASAH1, CATD, SCRB2, and TPP1 to *CLN1*, and the etiological links between NCLs drawn by these proteins, are intriguing areas of investigation that fall beyond the scope of this study.

## Palmitoylation–depalmitoylation cycles at the synapse

The PATs DHHC5 and DHHC17 have been described to function locally and continuously to palmitoylate synaptic substrates [14,66,67]. We documented a likely DHHC17-PPT1 partnership for palmitoylation–depalmitoylation cycles at the synapse (**S3 Fig**) by identifying 9

synaptic PPT1 substrates that contain the DHHC17 recognition motif ([VIAP][VIT]xxQP) [2]. These include CLH1, DYN1, SYNJ1, and DNJC5 (**S2 Table**), which have known functions in synaptic vesicle recycling [68–75]. Strikingly, previous studies have identified synaptic vesicle endocytosis deficits in PPT1 KO neurons [25,60] including reduced presynaptic vesicle pool size, deficits in evoked presynaptic vesicle release [23,25], and the persistence of VAMP2 and SNAP25 on the presynaptic membrane [23]. We hypothesize that palmitoylation–depalmitoylation dynamics, regulated by DHHC17 and PPT1, respectively, allow for efficient membrane association and dissociation of endocytic proteins during synaptic vesicle endocytosis. However, the exact timing and localization of depalmitoylation and the triggers for this cycle remain to be clarified. Identification of motifs recognized by DHHC5 may provide additional insights into local regulation of other PPT1 substrates.

## Depalmitoylation plays a role in disulfide bond formation and critical synaptic functions

Previous literature has established that palmitoylation of synaptic proteins is essential for their trafficking from the soma to the nerve terminal [76,77]. The palmitoylation of synaptic proteins destined for the secretory pathway is accomplished by resident PATs in the lumen of the endoplasmic reticulum or Golgi [48], while for other proteins, palmitoylation occurs on the cytosolic surface of these organelles. Although the precise PATs serving these distinct roles are not known, palmitoylation facilitates sorting (secretory) and hitching (cytosolic) of synaptic proteins to the synaptic vesicle precursor membrane [7,9]. Beyond keeping synaptically targeted proteins tethered to vesicles, we hypothesize that palmitoylation keeps synaptic proteins inert during axonal trafficking by preventing the formation of disulfide bonds. Hence, depalmitoylation by PPT1 may relieve brakes on ectopic adhesion interactions or enzymatic activity enforced by palmitate-mediated blockage of disulfide bonding. Once at the synapse, cytosolic proteins (such as G-proteins, endocytic proteins, kinases, and phosphatases) are released from the synaptic vesicle precursor, while membrane proteins (synaptic adhesion molecules, channels, and transporters) are inserted into the synaptic membrane and interact in both *cis* and *trans*. In the case of membrane proteins with extracellular palmitoylation sites, it is likely that depalmitoylated cysteines spontaneously form disulfide bonds in the oxidative extracellular environment, leading to functional maturation [48,78]. Our results suggest that local synaptic depalmitoylation by PPT1 modulates these functions, although they do not exclude PPT1-mediated depalmitoylation of substrates en route to the synapse. In PPT1 KO brains, we predict that palmitoylated synaptic proteins traffic correctly (as PATs are normal, and we observe few changes in the synaptic proteome; **Fig 2D**) but may function poorly at the synapse due to absent or compromised depalmitoylation (**Fig 2B**). In line with this hypothesis, we confirmed the synaptic localization of several validated PPT1 substrates in PPT1 KO by subcellular fractionation (**Fig 3D, S2B Fig**) and that PPT1 is found both in the synaptic cytosol (**S2B and S2D Fig**) and secreted from neurons (**Fig 4D**).

Our finding that PPT1 depalmitoylates IgG domains of cell adhesion molecules suggests an important role for PPT1 in regulating synaptogenesis (**Fig 4A–4C**). This finding is congruent with the impact of depalmitoylation on "synaptogenesis signaling" in PPT1 KO, identified by IPA (**Fig 3F**), as well as with previous findings on the importance of NRCAM palmitoylation for neuronal morphogenesis [79]. A *Drosophila* orthologue of NRCAM was also identified in a genetic modifier screen of fly PPT1 KO-induced degeneration [80]. Indeed, the aberrant palmitoylation of IgG class adhesion molecule substrates may explain postsynaptic deficits previously described in *CLN1* models, including immature dendritic spine morphology of cultured PPT1-null neurons [24,63]. Similarly, our finding that PPT1 depalmitoylates GluA1 and

several AMPAR interactors, and changes to "synaptic long-term depression" by IPA (**Fig 3F**), are congruent with LTP impairments reported in PPT1-deficient mice [24], and accumulating evidence that palmitoylation plays roles in modulating structural synaptic plasticity [81–83]. Since hyperpalmitoylation of known GluA1 palmitoylation sites can alter seizure susceptibility [52], our discovery of a novel palmitoylation site that is critical for AMPAR function (C323, **Fig 3E**) opens avenues for investigation of the molecular basis of epilepsy in *CLN1*.

We identified AT1B2, the β2 subunit of the $Na^+/K^+$ transporting ATPase heterodimer, as a high-confidence PPT1 substrate. AT1B2 contains 3 pairs of disulfide-bonded cysteines (validated by dNEM moieties) that exhibit aberrant palmitoylation in the PPT1 KO. These data strongly suggest that PPT1-mediated depalmitoylation facilitates the stabilization of β-sheet structures in this molecule through the regulation of disulfide bonds (**Fig 5G–5I**). Other $Na^+/K^+$ ATPase isoforms, AT1A1, AT1A2, AT1A3, and AT1B1, were also identified as high-confidence PPT1 substrates, accounting for all $Na^+/K^+$ ATPase subunits known to be expressed in neurons (**Fig 3C**) [84]. Similar to AT1B2, the β1 subunit (AT1B1) displays persistent palmitoylation in the PPT1 KO on a disulfide-bonded cysteine (C126; **S5 Table**). α subunits are not predicted to form disulfide bonds, but also exhibit excessive palmitoylation, as is expected of PPT1 substrates (**S5 Table**). AT1B2 and AT1A3 are known to form a complex that is essential for anchoring the protein retinoschisin (RS1) to plasma membranes. In the retina, this cell–adhesion interaction maintains retinal architecture and the photoreceptor bipolar cell synaptic structure [27,85,86]. Mutations in RS1 lead to juvenile retinoschisis, a form of early-onset macular degeneration [27,85,86]. Since juvenile retinal degeneration is also a feature of NCLs, these two diseases can be conflated and misdiagnosed in the clinic [87]. This is the first piece of molecular evidence, to our knowledge, that hints at an etiological overlap. Furthermore, basigin (BASI), which is essential for normal retinal development [88], was also found to be a PPT1 substrate and colocalizes with AT1B2 in the retina [89]. AT1B2 has also been shown to regulate neuron–astrocyte adhesion [84], although its role as a cell adhesion molecule in the brain is not fully characterized. $Na^+/K^+$ ATPase subunits are classified as Channels/Transporters (**Fig 3C**) but can also be considered cell adhesion molecules; high-confidence PPT1 substrates predominantly fall in this category, highlighting the importance of depalmitoylation in synaptic adhesion interactions.

## Conclusions

Identification of synaptic PPT1 substrates and the detection of PPT1 enzyme activity at synapses provide a rational basis for investigating fundamental mechanisms by which depalmitoylation regulates synaptic functions. As aberrant palmitoylation has been described in Huntington disease, Alzheimer disease, and amyotrophic lateral sclerosis (ALS), these data may have implications for multiple neurodegenerative disorders. This resource also establishes a detailed landscape of perturbations of depalmitoylation in NCL, opening avenues for molecular dissection of disease mechanisms.

## Methods

### Lead contact and materials availability

Sreeganga Chandra (sreeganga.chandra@yale.edu) is the lead contact for this paper. Mass spectrometry data are available through the ProteomeXchange Consortium in the PRIDE partner repository with accession PXD032052. Individual numerical source data and statistical analyses that underlie the summary data displayed in Figs 1D, 2B, 2D, 2E, 2F, 2H, 3E, 4D, 4F, 5C, 5D, 5E, 5F and 5G and S1B, S1D, S1E, S1F, S1H, and S2C, S2D, S2E Figs are available in **S1 Data**. Key materials and reagents are listed in **Table 3**.

**Table 3. Key reagents and resources.**

| REAGENT or RESOURCE | SOURCE | IDENTIFIER |
|---|---|---|
| **Antibodies** | | |
| Rabbit polyclonal anti-CSPα (used at 1:5,000) | Chemicon | Cat# AB1576; RRID: AB_90794 |
| Rabbit polyclonal anti-endophilin 1 (used at 1:1,000) | Synaptic Systems | Cat# 159 002; RRID: AB_887757 |
| Mouse monoclonal anti-SNAP-25 (used at 1:5,000) | Covance | Cat# SMI-81; RRID: AB_2315336 |
| Mouse monoclonal anti-β-Actin (used at 1:1,000) | GeneTex | Cat# GTX629630; RRID: AB_2728646 |
| Mouse monoclonal anti-dynamin (used at 1:2,000) | BD Biosciences | Cat# 610246; RRID: AB_397641 |
| Rabbit polyclonal anti-NrCAM (used at 1:10,000) | Abcam | Cat# ab24344; RRID: AB_448024 |
| Rabbit polyclonal anti-ATP1B2 (used at 1:200) | Thermo Fisher Scientific | Cat# PA5-11852; AB_2857261 |
| Rabbit polyclonal anti-GluA1 (used at 1:1,000) | Millipore | Cat# AB1504; RRID: AB_2113602 |
| Rat anti-Syncam2 (used at 1:2,000 for ICC) | Gift from the laboratory of Dr. Thomas Biederer | N/A–Custom Antibody |
| Chicken polyclonal anti-MAP2 (used at 1:500 for ICC) | Millipore | Cat# AB5543; RRID: AB_571049 |
| Mouse monoclonal anti-myelin basic protein (MBP) (used at 1:750) | Abcam | Cat# ab62631; RRID: AB_956157 |
| Rabbit polyclonal anti-synaptogyrin 3 (used at 1:1,000) | Synaptic Systems | Cat# 103 3030; RRID: AB_2619753 |
| Mouse monoclonal anti-synaptophysin 1 (used at 1:10,000 for WB, 1:500 for ICC) | Synaptic Systems | Cat# 101 011; RRID: AB_887822 |
| Rabbit monoclonal anti-α-Synuclein (used at 1:1,000) | Cell Signaling Technology | Cat# 4179; RRID: AB_1904156 |
| Rabbit polyclonal anti-PPT1 (used at 1:200 for ICC) | GeneTex | Cat #: GTX110677; RRID: AB_1951370 |
| Rabbit polyclonal anti-PPT1 (used at 1:50 for WB) | This paper; Thermo Fisher Scientific Life Sciences Solutions | Custom Antibody; Project VH2380 |
| IRDye 680RD goat anti-mouse IgG secondary | LI-COR Biosciences | Cat# 926–68070; RRID: AB_10956588 |
| IRDye 680RD goat anti-rabbit IgG secondary | LI-COR Biosciences | Cat# 926–68071: RRID: AB_10956166 |
| IRDye 800CW goat anti-rabbit IgG secondary | LI-COR Biosciences | Cat# 926–32211; RRID: AB_621843 |
| IRDye 800CW donkey anti-mouse IgG secondary | LI-COR Biosciences | Cat# 926–32212; RRID: AB_621847 |
| Goat anti-rat IgG (H+L) cross-adsorbed, Alexa Fluor 488 | Invitrogen | Cat# A11006; RRID: AB_141373 |
| Goat polyclonal anti-chicken IgY (H+L) cross-adsorbed, Alexa Fluor 633 secondary | Invitrogen | Cat# A21103; RRID: AB_2535756 |
| **Chemicals, peptides, and recombinant proteins** | | |
| TCEP | Thermo Fisher Scientific | Cat# 20490 |
| HA | Millipore Sigma | Cat# 467804 |
| NEM | Thermo Fisher Scientific | Cat# 23030 |
| Thiopropyl sepharose 4B | Millipore Sigma | Cat# T8512 |
| dNEM | Cambridge Isotope Laboratories | Cat# DLM-6711-10 |
| QuikChange II XL Site-Directed Mutagenesis Kit | Agilent | Cat# 200521 |
| 4-methylumbelliferyl-6-thio-Palmitate-β-D-glucopyranoside (MU-6S-Palm-βGlc) | Carbosynth | Cat# EM06650 |
| β-glucosidase from almonds | Sigma | Cat# 49290 |
| **Deposited data** | | |
| Analyzed mass spectrometry data | This paper | Tables |
| Raw data | ProteomeXchange Consortium | PXD017270 |
| **Experimental models: Cell lines** | | |
| HEK 293T cells | American Type Culture Collection | Cat# CRL-3216 |
| **Experimental models: Organisms/strains** | | |
| *Mus musculus*—B6;129-Ppt1tm1Hof/J | The Jackson Labs | JAX: 004313 |
| *Xenopus laevis* | Nasco | LM00531 |
| **Recombinant DNA** | | |
| LCV-hPPT1 | This paper | N/A |
| LCV-FUGW | This paper | N/A |

(*Continued*)

**Table 3.** (Continued)

| REAGENT or RESOURCE | SOURCE | IDENTIFIER |
|---|---|---|
| pGEMHE-GluA1.WT | This paper | N/A |
| pGEMHE-GluA1.C323A | This paper | N/A |
| pGEMHE-TARPg8 | This paper | N/A |
| **Software and algorithms** | | |
| CSS Palm | Ren and colleagues (2008) | http://csspalm.biocuckoo.org |
| Swiss Palm | Blanc and colleagues (2015) | https://swisspalm.org |
| IPA | Qiagen | https://www.qiagenbioinformatics.com/products/ingenuity-pathway-analysis |
| GraphPad Prism 7 | GraphPad | https://www.graphpad.com/scientific-software/prism |
| Progenesis QI Proteomics software V4.0 | Nonlinear Dynamics | http://www.nonlinear.com/progenesis/qi-for-proteomics/download |
| Mascot search algorithm | Matrix Science | http://www.matrixscience.com/search_form_select.html |
| WebLogo | Crooks and colleagues (2004) | https://weblogo.berkeley.edu |
| SWISS-MODEL | Bienert and colleagues (2017) | https://swissmodel.expasy.org |

Sourcing and catalog information for materials and resources used to conduct this study.

dNEM, $d_5$-$N$-ethylmaleimide; HA, hydroxylamine; IPA, Ingenuity Pathway Analysis; NEM, N-ethylmaleimide; TCEP, tris(2-carboxyethyl)phosphine.

## Experimental model and subject details

PPT1 KO (B6;129-Ppt1tm1Hof/J) and WT (C57BL6/J) mice were obtained from The Jackson Labs (Bar Harbor, ME). PPT1 KO mice were originally generated by targeted genetic disruption of exon 9 of the *Ppt1* gene [22]. The PPT1 KO line was backcrossed to C57BL6/J to obtain a homogenous genetic background. The PPT1 KO line was maintained by heterozygous breedings. Homozygous PPT1 KO and WT littermate controls were used for key synaptosome proteomics experiments that established putative PPT1 substrates. Age-matched C57BL6/J WT animals were used as controls in succeeding experiments. Mice were 2 months of age and of both sexes. Sex differences were not measured because sexes were equally mixed in synaptosome experiments. Animal care and housing complied with the Guide for the Care and Use of Laboratory Animals [90] and were provided by the Yale Animal Resource Center (YARC). Animals were maintained in a 12-hour light/dark cycle with ad libitum access to food and water. Female *X. laevis* frogs were obtained from Nasco (Fort Atkinson, WI) for use in oocyte preparation. All experimental protocols involving animals were approved by the Institutional Animal Care & Use Committee (IACUC) at Yale University (Protocols 2021–11117 and 2020–11029).

HEK 293T cells were obtained from American Type Culture Collection (ATCC).

## Method details

**Preparation of synaptosomes.** Forebrains were freshly harvested from 2 mice (age = 2 months), suspended in ice cold Buffer A (320 mM sucrose, 10 mM HEPES, pH 7.4 with protease inhibitors), then homogenized in 12 up-down passes at 900 rpm in a glass Teflon homogenizer. Homogenate (total) was centrifuged at 800 g for 10 minutes, 4˚C. The supernatant (S1) was then centrifuged at 9,000 g for 15 minutes, 4˚C. The resulting pellet was resuspended in 3 mL ice cold Buffer A and centrifuged at 9,000 g for 15 minutes, 4˚C to obtain the synaptosomal supernatant (S2) and washed synaptosomes (P2') for Acyl RAC and PPT1 validation.

**Synaptic fractionation.** Following synaptosome preparation from WT and PPT1 KO mouse whole brains (*n* = 8 per genotype; age = 2 months), synaptic fractions were prepared as previously described [91]. Briefly, synaptosomes were lysed by hypoosmotic shock and the

lysis supernatant (LS1) and synaptosomal membrane pellet (LP1) were collected. LP1 was resuspended and LS1 was centrifuged at 100,000 g, 4˚C for 1 hour to obtain synaptic vesicles (LP2) and synaptic cytosol (LS2). LP1 was then subjected to sucrose density gradient centrifugation at 48,000 g, 4˚ C for 2.5 hours to obtain the myelin fraction (MF), SPM, and mitochondrial fraction (Mito.). The purity of subcellular fractions was assessed through enrichment of markers of synaptic subcompartments as determined by quantitative immunoblotting: synaptogyrin 3 (SNG3), synaptophysin 1 (SYPH), and synaptobrevin 2 (VAMP2) in LP2; α-synuclein (SYUA) in LS2; myelin basic protein (MBP) in MF.

**Acyl RAC.** Acyl RAC was performed essentially as described in Henderson and colleagues [29]. Briefly, homogenized whole brains or prepared synaptosomes were blocked in 10 mM NEM, then subjected to chloroform-methanol precipitation (Chloroform:MeOH:water:: 1:4:3 volumes) to remove free NEM. Samples were resuspended in 2% SDS, 50 mM Tris, 5 mM EDTA, pH 7.0 with protease inhibitors and 10 mM TCEP and resolublized for 20 to 40 minutes at 37˚C. Samples were then incubated at 4˚C overnight with 10 mM NEM. Two additional chloroform-methanol precipitations were performed. Protein was resuspended in 2% SDS, 50 mM Tris, 5 mM EDTA, pH 7.4 and incubated at 37˚C until complete dissolution. Samples were diluted 1:10 with 150 mM NaCl, 50 mM Tris, 5 mM EDTA, pH 7.4 with 0.2% Triton X-100, 1 mM PMSF, and protease inhibitor cocktail following dissolution. Another chloroform methanol precipitation was performed, and the protein precipitate was dissolved in 4% SDS in Buffer A (100 mM HEPES, 1 mM EDTA, pH 7.5) at 37˚C. Following dissolution, the protein sample was diluted to 2% SDS in Buffer A and split into + HA and control samples. At this step, samples for total protein expression were collected for mass spectrometry. HA was added to the paired samples at a total concentration of 500 mM at neutral pH and incubated at 4˚C overnight. Thiopropyl sepharose beads were prewashed 4 times in water, incubated in Buffer B (1% SDS, 100mM HEPES, 1mM EDTA, pH 7.5), then incubated with protein samples for 2 hours at room temperature. The beads were washed 5 times with Buffer B to remove nonspecific contaminants, followed by Buffer B with 50 mM DTT to elute previously palmitoylated protein from the beads. Beads were incubated for 20 minutes at room temperature before supernatant was obtained for mass spectrometric analysis.

**Mass spectrometry.** Proteomics analyses were performed and analyzed at the Yale Mass Spectrometry (MS) and Proteomics Resource of the W.M. Keck Foundation Biotechnology Resource Laboratory with the support of the Yale/NIDA Neuroproteomics Center. Brain homogenates or synaptosomes were lysed with RIPA buffer containing protease and phosphatase inhibitor cocktail using ultra-sonication. Cellular debris were removed by centrifugation at 16,000g for 10 minutes at 4˚C. A total of 150 μL of the supernatant was transferred to a new tube and proteins were precipitated with Chloroform:MeOH:water (100:400:300 μL). The protein pellet was washed 3 times with cold methanol prior to air drying for 5 minutes. Dried protein pellets were resolubilized with 8 M urea containing 400 mM ammonium bicarbonate. Protein concentration was measured using Nanodrop (Thermo Fisher Scientific, Waltham, Massachusetts, USA), and 100 μg of each sample was taken for additional downstream sample preparation. The cysteines within the 100-μg protein samples were reduced with DTT at 37˚C for 30 minutes and alkylated with iodoacetamide at room temperature in the dark for 30 minutes. Reduced and alkylated proteins were then digested with Lys-C (1:25 enzyme:protein ratio) overnight, and subsequently with trypsin (1:25 enzyme:protein ratio) for 7 hours at 37˚C. Digestion was quenched with 20% trifluoroacetic acid, and samples were desalted using C18 reverse phase macrospin columns (The Nest Group, Southborough, Massachusetts, USA). Then, eluted peptides were dried using SpeedVac. Total peptide concentration was determined by nanodrop after reconstitution with 0.1% formic in water. Dilutions were made to ensure that equal total amounts (0.25 μg) were loaded on the column for Liquid Chromatography

MS/MS analyses. Retention Calibration Mix (Thermo Fisher Scientific) was added equally to all the peptide solutions to be analyzed within the set of comparative samples for system quality control and downstream normalization. LFQ of protein samples was performed on a mass spectrometer (Thermo Scientific Orbitrap Fusion or Thermo Scientific Q-Exactive Plus) connected to a UPLC system (Waters nanoACQUITY) equipped with a Waters Symmetry C18 180 μm × 20 mm trap column and a 1.7-μm, 75 μm × 250 mm nano ACQUITY UPLC column (35°C). Additional details on UPLC and mass spectrometer conditions can be found in Charkoftaki and colleagues 2019 [92]. The LC-MS/MS data was processed using Progenesis QI Proteomics software (Nonlinear Dynamics, version 4.0), and protein identification was carried out using the Mascot search algorithm (Matrix Science, Boston, Massachusetts, USA).

**Pathway analysis.** Pathway analyses were conducted with 3 distinct software platforms— IPA available from Qiagen (Redwood City, CA), Gene Ontology Resource, and STRING analysis.

**PPT1 validation.** HEK293T cells were plated at $2 \times 10^6$ cells per plate on 10 cm$^2$ dishes 1 hour prior to transfection with LCV-mPPT1 or LCV-FUGW using GenePorter3000. After 48 hours of incubation, growth media was removed from the cells and filtered through a 0.22-μm filter. Media was then concentrated using a 10,000 NMWL centrifugal filter at 2,900 g at 4°C for a total of 40 minutes. Concentrated media was then brought to 5 mM MgCl$_2$ and 10 mM Tris HCl, pH 7.2 and protease inhibitors were added. The Acyl RAC procedure was performed as above, except synaptosomes were resuspended at pH 5.5 and divided into 3 parts prior to HA incubation. PPT1 media, GFP media, or HA was then added to synaptosomes during the HA step of Acyl RAC to depalmitoylate proteins. The samples were then subjected to LFQ mass spectrometry as above.

**Generation of custom PPT1 antibody.** Polyclonal antibodies were raised in rabbit against a PPT1 peptide antigen (STLYTEDRLGLKKMDKAGK), and serum was affinity purified (Thermo Fisher Life Custom Antibodies, Life Sciences Solutions). Antibody specificity was validated with PPT1 KO whole brain homogenates.

**Western blot analyses.** SDS-PAGE and western blots were performed using standard procedures. Images and densitometry values (where applicable) were collected using a LI-COR Odyssey imaging system. Western blot signal quantifications are represented as mean ± SEM.

**ICC and confocal microscopy.** Primary hippocampal neurons from WT and PPT1 KO mice (P0) were cultured on coverslips, as previously described [93]. For lentiviral transductions, lentivirus particles were added to the media at DIV4. At DIV 14, neurons were fixed with 4% buffered paraformaldehyde (PFA) with 4% sucrose, washed in 1X PBS, and blocked in 3% goat serum at room temperature. Neurons were incubated in primary antibodies overnight at 4°C, then in Alexa-conjugated secondaries at 4°C. Coverslips were prepared in antifade mounting medium with DAPI (H-1000 Vectashield) and sealed. Fluorescent images were collected with a Zeiss LSM 800 laser scanning confocal microscope (Oberkochen, Germany) with a 63× oil immersion objective.

**Plasmid construction and lentiviral preparation.** GluA1 Cysteine 323 to Alanine (GluA1.C323A) was generated using QuikChange site-directed mutagenesis (Agilent, Santa Clara, CA). GluA1.WT, GluA1.C323A, and TARPγ8 were subcloned into pGEMHE (Addgene, Watertown, MA). Lentiviruses for hPPT1 were prepared by transfecting HEK293T cells with LCV-hPPT1 and viral constructs. Virus was collected from the media, concentrated, and stored at −80 degrees until use.

***Xenopus laevis* oocyte electrophysiology.** Oocytes were surgically extracted from anesthetized *X. laevis* frogs and defolliculated with collagenase, as previously described [94]. Two-electrode voltage clamp (TEVC) recordings were then performed as previously described [95]. Briefly, cRNAs were transcribed in vitro with a T7 mMessage mMachine (Ambion, Austin, TX). Manually selected *X. laevis* oocytes were injected with cRNA of GluA1.WT or GluA1.

C323A (100 pg), either with TARPγ8 (100 pg), or alone at a higher concentration (2 ng). TEVC recordings ($E_h = -70$ mV) were taken 2 days after injection at room temperature. Glutamate (1 mM) was bath applied in recording solution (90 mM NaCl, 1.0 mM KCl, 1.5 mM CaCl$_2$, 10 mM HEPES, pH 7.4) with cyclothiazide (50 μM) to block desensitization of AMPARs. Data are represented as mean ± SEM.

**PPT1 enzyme activity.**   To test secreted PPT1 activity, media were collected from WT and PPT1 KO primary hippocampal neuron cultures at DIV 14, passed through a 0.22 μM Millex-GV filter unit, and concentrated 5X with an Amicon centrifugal filter unit with a 10-kDa cutoff (Millipore Sigma, Burlington, MA).

Concentrated media or subcellular fractions were thawed on ice. PPT1 enzyme activity was determined by the standard 1-step assay described by van Diggelen and colleagues 1999 [96]. Briefly, 10 μg (in 10 μL) of each sample was incubated with 20 μL of PPT1 substrate solution (0.64 mM 4-methylumbelliferyl-6-thio-Palmitate-β-D-glucopyranoside (MU-6S-Palm-βGlc), 15 mM DTT, 0.375% (w/v) Triton X-100, and 0.1 U β-glucosidase from almonds in McIlvain's phosphate/citrate buffer, pH 4.0) or control solution without MU-6S-Palm-βGlc for 1 hour at 37˚C. The reaction was terminated with the addition of 200 μL 0.5 M NaHCO$_3$/0.5 M Na$_2$CO$_3$, pH 10.5 with 0.025% (w/v) Triton X-100. Released 4-methylumbelliferone (ME) fluorescence was measured in technical triplicate with a BioTek Synergy H1 fluorometer (excitation 380 nm, emission 454 nm) and normalized to substrate-free controls.

**Quantification and statistical analysis.**   MS protein data were normalized by comparing the abundance of the spike in Pierce Retention Time Calibration mixture among all the samples as well as the sum of squares of spectral count for each replicate. Relative protein-level fold changes were calculated from the sum of all unique, normalized peptide ion abundances for each protein on each run. Only proteins identified by at least 2 unique peptides were considered. Mass spectrometry data *p*-values were calculated using a 2-tailed *t* test (*n* = 3 biological, 3 technical replicates; *n* = 9 total per genotype per experiment). While this does not meet the *t* test condition for independence, we proceeded to maximize our hits. We also did not use FDR correction for multiple comparisons in our screen to maximize our hits. In addition, we used 1.5-fold up- or down-regulation as cutoffs for the mass spectrometric data. Peptide-level data are represented as percentages or ratios of summed normalized peptide ion abundances, as described in figure legends. Peptide quantifications are represented as mean ± SD.

ICC image analysis was executed in FIJI and performed blinded to genotype. Files were converted to 8-bit images and 50 μM dendritic ROIs were selected for analysis. The "analyze particles" function was used to obtain puncta number and average puncta size for puncta larger than 20-pixel units (mean gray value scale: CADM2, 100–255; SYPH, 75–255). To assess colocalization, Pearson correlation coefficient was calculated using the "Coloc 2" function. Data are represented as mean ± SD.

Throughout, *p*-values were calculated using 2-tailed *t* tests, and *p*-values <0.05 were considered significant. All protein and gene names are listed using UniProt nomenclature. Figures were generated in GraphPad Prism 7.0, Adobe Illustrator, and IPA.

## Supporting information

**S1 Data. Excel spreadsheet containing individual numerical source data and statistical analyses that underlie the summary data displayed in Figs 1D, 2B, 2D, 2E, 2F, 2H, 3E, 4D, 4F, 5C, 5D, 5E, 5F and 5G and S1B, S1D, S1E, S1F, S1H, S2C, S2D and S2E Figs.** (XLSX)

**S1 Table. WT and PPT1 KO synaptic protein expression.** Average KO/WT protein expression ratios for data shown in **Fig 2D**. *p*-Values were calculated using a 2-tailed *t* test. Blue

denotes decreases (<1.5 fold), red denotes increases (>1.5 fold), while black denotes unchanged expression. KO, knockout; PPT1, palmitoyl protein thioesterase 1; WT, wild-type.
(PDF)

**S2 Table. Proteins exhibiting significant changes in WT versus PPT1 KO synaptic palmitome.** These data correspond to **Fig 2B** and are the expression ratios for proteins that exhibit significant changes in the palmitome ($n = 242$). Proteins marked with an asterisk (*) were removed from subsequent consideration as PPT1 substrates due to decreased palmitoylation ($n = 4$: HDHD2, S1PR1, SNX1, UBQL2; blue <1.5-fold change), presence in the CRAPome [97] ($n = 9$: 1433E, COF1, HNRPU, KPYM, RL13, RL22, RL4, RLA0, TCPD), lack of a cysteine residue ($n = 2$: NDUA6, SYUA), increased palmitoylation and protein expression ($n = 4$: ASAH1, CATD, SCRB2, and TPP1), or lack of detection in the synaptic proteome ($n = 19$: CKAP4, CPNE1, FHL1, GPC5B, ITM2B, ITM2C, LGI2, MAGI1, NSMA2, PP1G, PRDX4, R7BP, RB3GP, S39AC, TPPC3, VAMP7, XKR4, S1PR1, MYO6, PDPR; removed proteins S1PR1, HNRPU, and RL22 also fall in this category). The remaining 204 proteins are the final list of putative PPT1 substrates prior to the validation screen. *p*-Values were calculated using a 2-tailed *t* test. Red denotes significantly increased expression in the palmitome (>1.5-fold). KO, knockout; PPT1, palmitoyl protein thioesterase 1; WT, wild-type.
(PDF)

**S3 Table. Formerly palmitoylated peptides identified for putative PPT1 substrates.** [x] Carbamidomethyl indicates location of palmitoylated cysteine (C) in peptide sequence. PPT1, palmitoyl protein thioesterase 1.
(PDF)

**S4 Table. Residual PPT1 substrates.** Proteins identified in the secondary validation screen that were not identified as putative substrates in the primary screen. PPT1, palmitoyl protein thioesterase 1.
(PDF)

**S5 Table. Carbamidomethyl-modified peptides for high-confidence PPT1 substrates.** Protein localization and disulfide bond data were gathered from the UniProt entry for each protein. Carbamidomethyl (carba.) sites that were not validated in the tertiary screen are indicated by an asterisk (*). Disulfide bonds validated by a dNEM moiety in the tertiary screen are indicated (X). GPI, GPi anchor; MM, mitochondrial membrane; SPM, synaptic plasma membrane; PPT1, palmitoyl protein thioesterase 1.
(PDF)

**S1 Fig. Generation of WT and PPT1 KO palmitome and proteome expands the repertoire of known palmitoylated proteins in the brain.** **(A)** Venn diagram of 3 independent palmitome experiments exhibits 1,795 common proteins. **(B)** Volcano plot of fold change between genotypes (PPT1 KO/WT) for common proteins with putative synaptic PPT1 substrates in blue (**Fig 2B**, **S2 Table**, **S1 Data**). A total of 15 palmitoylated proteins are significantly differentially expressed in whole brain (3 decreased, including PPT1; 12 increased; 1.5-fold, $p < 0.05$; blue lines). **(C)** Venn diagram of 3 independent proteome experiments exhibits 1,873 common proteins. **(D)** Volcano plot of fold change between genotypes (KO/WT) for common proteins with putative synaptic PPT1 substrates in blue (**Fig 2B**, **S2 Table**, **S1 Data**). A total of 5 proteins are significantly up-regulated (1.5-fold, $p < 0.05$; blue lines). Other depalmitoylating enzymes (green points) do not display compensatory up-regulation of protein expression. **(E)** Palmitoylated protein expression was highly correlated between genotypes for palmitome hits ($m = 0.9753 \pm 0.0030$; $R^2 = 0.9857$; **S1 Data**). **(F)** Protein expression was highly correlated

between genotypes for proteome hits (m = 0.8675 ± 0.0037; $R^2$ = 0.7711). When the MBP outlier was removed, this correlation was improved (m = 0.9802 ± 0.0023; $R^2$ = 0.9916; **S1 Data**). Red lines indicate 1:1 WT to PPT1 KO protein expression ratio. **(G)** Venn diagram of 1,290 proteins common between whole brain proteome and palmitome (*n* = 3 each). **(H)** Protein expression levels compared to palmitoylation levels for significantly changed proteins in palmitome. Proteins in orange region are significantly increased (1.5-fold; *p* < 0.05) in both the proteome and palmitome. Proteins in blue region display decreased or unchanged protein expression and increased palmitoylation. Red line indicates equal expression and palmitoylation levels (x = y; **S1 Data**). KO, knockout; PPT1, palmitoyl protein thioesterase 1; WT, wild-type.
(TIF)

**S2 Fig. PPT1 is expressed and enzymatically active in synaptic cytosolic subcompartments in neurons. (A)** hPPT1 (green) colocalizes with synaptophysin 1 (SYPH; red) in MAP2+ neurites in WT mouse primary neuronal culture (50 μm scale bars; 10 μm scale bar in higher magnification image). Lentiviral transduction of human PPT1 was performed due to lack of commercially available antibodies that can detect mouse PPT1 by ICC (including the custom antibody used for western blotting). **(B)** Endogenous PPT1 is present in all synaptic fractions of a subcellular fractionation of whole brains (8 WT and 8 PPT1 KO mice; age = 2 months; run in biological triplicate) with the highest level in synaptic cytosol (LS2). **(C)** Immunoblot quantifications of 3 subcellular fractionation experiments (**S1 Data**). Markers of synaptic subcompartments are appropriately localized and enriched (fold change from total) for both genotypes (WT, red; PPT1 KO, black), with no difference between genotypes: SNG3, synaptophysin 1 (SYPH), and synaptobrevin 2 (VAMP2) in synaptic vesicle enriched fraction (LP2); α-synuclein (SYUA) in synaptic cytosolic fraction (LS2); MBP in MF. **(D)** WT synaptic subcompartment fractions display significant PPT1 enzymatic activity, most notably the synaptic cytosol (LS2). Negligible enzymatic activity was detected in PPT1 KO total and P2' fractions shown for comparison. Bars represent the mean of 3 technical and 3 biological replicates normalized to substrate-free controls, with SEM error bars (* *p* < 0.05; ** *p* < 0.01; **S1 Data**). **(E)** Average PPT1 protein concentration determined by quantitative immunoblotting (B) is highly correlated with average PPT1 enzymatic activity (D) (y = 28565x – 12.61; $R^2$ = 0.8291; **S1 Data**). ICC, immunocytochemistry; KO, knockout; LS1, LP1, crude synaptic cytosol and membrane; LP2, synaptic vesicles; LS2, synaptic cytosol; MBP, myelin basic protein; MF, myelin fraction; Mito, mitochondria; P2', crude synaptosomes; PPT1, palmitoyl protein thioesterase 1; S1, first supernatant; S2, second supernatant; SNG3, synaptogyrin 3; SPM, synaptic plasma membrane; WT, wild-type.
(TIF)

**S3 Fig. Endocytic PPT1 substrates contain the DHHC17 recognition motif. (A)** Alignment of DHHC17 recognition motif for 9 endocytic PPT1 substrates with 26 amino acids on either side. The motif is noted above the logo plot. There are no other motifs surrounding the DHHC17 motif. **(B)** IPA of DHHC17 motif-containing proteins identifies clathrin-mediated endocytosis signaling as the top enriched pathway. These pathways account for 6 of the 9 motif-containing proteins. PPT1, palmitoyl protein thioesterase 1.
(TIF)

## Acknowledgments

We would like to thank Weiwei Wang and Jean Kanyo for mass spectrometry sample preparation and data collection; Na Wang for mutagenesis of GluA1 construct; and John E. Lee, Angus Nairn, and Arthur Horwich for reading and editing the manuscript.

## Author Contributions

**Conceptualization:** Erica L. Gorenberg, Sofia Massaro Tieze, Sreeganga S. Chandra.

**Data curation:** Erica L. Gorenberg, Sofia Massaro Tieze, TuKiet T. Lam.

**Formal analysis:** Erica L. Gorenberg, Sofia Massaro Tieze, Betül Yücel.

**Funding acquisition:** TuKiet T. Lam, Sreeganga S. Chandra.

**Investigation:** Erica L. Gorenberg, Sofia Massaro Tieze, Betül Yücel, Helen R. Zhao, Vicky Chou, Gregory S. Wirak, Susumu Tomita, TuKiet T. Lam.

**Methodology:** Erica L. Gorenberg, Sofia Massaro Tieze, Sreeganga S. Chandra.

**Resources:** TuKiet T. Lam.

**Supervision:** Sreeganga S. Chandra.

**Validation:** Erica L. Gorenberg, Sofia Massaro Tieze.

**Visualization:** Erica L. Gorenberg, Sofia Massaro Tieze, Betül Yücel.

**Writing – original draft:** Erica L. Gorenberg, Sofia Massaro Tieze, Sreeganga S. Chandra.

**Writing – review & editing:** Erica L. Gorenberg, Sofia Massaro Tieze, Betül Yücel, Helen R. Zhao, Vicky Chou, Gregory S. Wirak, Susumu Tomita, TuKiet T. Lam, Sreeganga S. Chandra.

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
