## [Editor Report · Decision Letter 0]

28 Sep 2021

Dear Dr Chandra, 

Thank you for submitting your manuscript entitled "Identification of PPT1 substrates highlights roles of depalmitoylation in disulfide bond formation and synaptic function" for consideration as a Methods and Resources Article by PLOS Biology.

Your manuscript has now been evaluated by the PLOS Biology editorial staff, as well as by an academic editor with relevant expertise, and I am writing to let you know that we would like to send your submission out for external peer review.

Please re-submit your manuscript within two working days, i.e. by Sep 30 2021 11:59PM.

Kind regards,

Gabriel Gasque

Senior Editor

PLOS Biology

ggasque@plos.org

---

## [Decision Letter · Decision Letter 1]

8 Nov 2021

Dear Dr Chandra,

Thank you for submitting your manuscript "Identification of PPT1 substrates highlights roles of depalmitoylation in disulfide bond formation and synaptic function" as a Methods and Resources article for review by PLOS Biology. Your paper was assessed and discussed by the PLOS Biology editors, by an academic editor with relevant expertise, and by four independent reviewers. Based on the reviews, I regret that we will not be pursuing this manuscript for publication in the journal. Please accept my apologies for the delay in sending this decision to you.

As you will see, while the reviewers agree that a systematic analysis of PPT1 substrates would be important and significant for the field, they raise a series of technical concerns that weaken the significance of your findings. Particularly, reviewer 2 notes that your experiments did not control for genetic background. Upon further discussion with the reviewers, reviewer 4 considered this flaw seriously undermines the conclusiveness of the data and suggested we rejected the study. Because we think this is a very serious concern, I am afraid that we must decline further consideration.

The reviews are attached, and we hope they may help you should you decide to revise the manuscript for submission elsewhere. I am sorry that we cannot be more positive on this occasion. 

I hope you appreciate the reasons for this decision and will consider PLOS Biology for other submissions in the future. Thank you for your support of PLOS and of Open Access publishing.

Sincerely,

Gabriel Gasque

Senior Editor

PLOS Biology

ggasque@plos.org

Reviewer remarks:

Reviewer #1: Gorenberg et al performed the acyl-RAC assay to systematically identify S-palmitoylated proteins from wild-type and PPT1 knockout mouse brains. They identified and validated PPT1 substrates in the brain, which include CSPa, Goa, NRCAM, CADM2, GluA1 and so on. Furthermore, the authors propose that depalmitoylation of transmembrane PPT1 substrates regulates their disulfide bond formation. Thus, this paper first provided a useful resource of global palmitoylated substrates of PPT1. Addressing the following points would strengthen this paper.

1. In Fig. 1C, SNAP-25 levels seems similar between +hydroxylamine (HA) and -HA samples. Because SNAP-25 is a representative palmitoylated protein, I am concerned about the specificity of the acyl-RAC assay.

2. Identified cysteine depalmitoylation sites by PPT1 include extracellular cysteines (Figs. 3E and 4A) and cytosolic ones (e.g, CSPa, Goa, dynamin-1), suggesting that PPT1 acts at the lumen side of ER and cytosol side. To clarify it, the authors should show the endogenous PPT1 localization using knockout-validated antibody or tagged knock-in approaches like SLENDR (Mikuni et al, Cell 2016) and ORANGE (Willems et al, PLOS BIOL 2020). 

3. Figure numbers of Fig. S2 are not correct. Please carefully check the manuscript. 

Reviewer #2: Infantile neuronal ceroid lipofuscinosis (INCL) is a uniformly fatal neurodegenerative lysosomal storage disease (LSD) caused by inactivating mutations in the CLN1 gene. CLN1 encodes palmitoyl-protein thioesterase-1 (PPT1), a lysosomal depalmitoylating enzyme. It has been proposed that PPT1-deficiency leads to lysosomal accumulation of S-palmitoylated proteins (constituents of ceroid) leading to INCL pathogenesis. Despite the discovery in 1995, that inactivating mutations in the CLN1 gene cause INCL, the substrates of PPT1 that accumulate in the lysosome and other organelles have remained unidentified, not for a lack of trying. In this manuscript, Gorenberg and colleagues used mass spectrometric analyses of proteins in synaptosomes purified from the brain tissues of WT and Cln1-/- mice and identify the putative substrates of PPT1. The identification of the substrates of this enzyme may unravel the pathogenic mechanism(s) underlying INCL and would be a major advance in this area of research. From this standpoint, the study of Gorenberg and colleagues attempt to provide important information, which, if validated, may help us to understand the mechanism of this devastating disease. While at the outset it appears to be an important study, some important questions need to be addressed.

Major points-

a. Using unbiased proteomic approaches, the authors claim to have identified 9 distinct classes of PPT1 substrates in synaptosomes purified from WT and Cln1-/- mouse brain. They show that Ppt1-deficiency in Cln1-/- mouse brain causes these substrates to accumulate in synaptosomes. Notably, the authors chose to use purified synaptosomes from the brain of 129 Cln1-/- mice and to identify the proteins accumulated in synaptosomes from the brain of these mice. Since genetic background may influence gene expression, it is prudent to use mice with identical genetic background. This is the reason why many investigators have first generated 129 Cln1-/- mice because the ES cells used to target the Cln1 gene was derived from 129 mice (See Gupta et al. PNAS 2001). In subsequent studies, the genetic background was converted to C57 by backcrossing 129 Cln1-/- mice with C57 WT mice >10 times (see the Methods in Dearborn, J.T. et al. Sci Rep. 5, 12752, 2015). The WT littermates from C57 Cln1-/- mice provided a homogeneous C57 genetic background. Alternatively, the authors could have used purified synaptosomes from 129 Cln1-/- and their WT littermates which would have obviated the time-consuming backcrosses to obtain C57 genetic background for both WT and KO mice. This way, the results of their proteomic studies comparing the putative Ppt1-substrates in synaptosomal preparations would have been on a solid ground. 

b. The main organelle in which PPT1 (CLN1 gene product) is localized in cells has been clearly established to be the lysosome (Verkruyse and Hofmann. JBC. 271,15831-15836, 1996; Hellsten et al. EMBO J. 15,5240-5245, 1996) although trace amounts of PPT1 have also been reported in extra lysosomal sites, like the synaptosomes. This may be because a small % of the soluble lysosomal proteins are known to be secreted instead of being targeted to the lysosome (Ballabio & Gieselmann. Biochim Biophys Acta. 1793, 684-696, 2009). Moreover, the major pathological features of INCL include the accumulation of S-palmitoylated (S-acylated) proteins in the lysosome. Further, the S-acylated proteins are the major constituents of ceroid (called granular osmiophilic deposits or GRODS) (Galvin, N. et al. Pediatr Dev Pathol. 11, 185-192, 2008). This is a characteristic pathological finding in the brain of INCL patients and in that of the Cln1-/- mice. Furthermore, it has been reported that the S-palmitoylated proteins require depalmitoylation for their degradation by lysosomal acid hydrolases (Lu, J.Y. & Hofmann, S.L. J Lipid Res. 47, 1352-1357, 2006). The authors' suggestion that "Protein degradation does not require depalmitoylation by PPT1" is premature. An explanation is needed as to why intravenous administration of high-dose PPT1-enzyme to Cln1-/- mice reduces lysosomal storage of S-palmitoylated proteins and modestly prolongs survival in a preclinical mouse model of INCL (Hu, J. et al. Mol Genet Metab. 107, 213-221, 2012). 

c. Since Ppt1 functions in an acidic environment of the lysosome, it would be important to present data showing: (i) the pH within the synaptosomes of WT and Cln1-/- mice and (ii) whether Ppt1 is enzymatically active in the synaptosomes from WT mice? In this regard, a very good assay is commercially available to evaluate the enzymatic activity of PPT1 (Van Diggelen, O.P. et al. Mol Genet Metab 66, 240-244, 1999). Do S-palmitoylated proteins accumulate in the synaptosomes of Cln1-/- mice? The reason for determining PPT1 enzymatic activity in synaptosomes is that if in WT synaptosomes PPT1 activity cannot be detected then the identification of putative PPT1 substrates in synaptosomes may not have any relevance to the disease. Since most of the PPT1 is localized in the lysosome and the method for purification of lysosomes from brain tissues is straight forward, the authors could identify the substrates of this enzyme using purified lysosomes. If the authors can confirm that the same 9 distinct classes of Ppt1-substrates, which they identified in the synaptosomes, also accumulate in the lysosomes from Cln1-/- mice, their proteomic data will be much more solid. 

d. In the methods section of the manuscript, I could not find how the authors determined the purity of the synaptosome preparations? Purification of the intracellular organelles is difficult, especially from the brain and the authors should describe the methodology is detail. 

e. The authors claim that they have identified ">100 novel PPT1 substrates". To confirm this finding, they used in vitro assays using recombinant PPT1 to demonstrate that those proteins are the substrates of Ppt1. This assumption may not be totally correct. For example, in in vitro assays the H-Ras protein has been reported to be depalmitoylated by PPT1 (Lu, J.Y. & Hofmann, S.L. J Biol Chem. 270,7251-7256,1995), whereas the enzyme that catalyzes the depalmitoylation of H-Ras in vivo is a cytosolic thioesterase, acyl-protein thioesterase-1 (APT1) (Duncan & Gilman J Biol Chem. 273,15830-7, 1998). For this reason alone, the authors could have used the proteins from purified lysosomes from Cln1-/- mouse brain and those from their WT littermates to authenticate the results from the synaptosomes. 

f. Auto acylation (auto palmitoylation) and depalmitoylation may occur spontaneously without palmitoyl acyltransferases (called ZDHHCs) and palmitoyl-protein thioesterases, respectively. It has been suggested that auto acylation plays important roles in the dynamic thioesterification of some cellular proteins like the G Protein a-subunits (Duncan & Gilman JBC 271, 23594-23600, 1996). During the rigorous procedure of synaptosome isolation and purification some proteins may undergo auto acylation when under in vivo conditions non-enzymatic palmitoylation-depalmitoylation may occur spontaneously, albeit at a very low level. Furthermore, constitutive deacylation/reacylation cycle operates on S-palmitoylated proteins. Thus, just by comparing the level of S-palmitoylated proteins in synaptosome preparations from WT and Cln1-/- mice may not identify the specific substrates of PPT1.

g. The authors claim that the "Identification of PPT1 substrates highlight roles of depalmitoylation in disulfide bond formation and synaptic function". Although it is a novel idea, I fail to understand the rationale and the validity of this statement. Disulfide bond formation in vivo primarily occurs within the endoplasmic reticulum (ER). It is catalyzed by a variety of oxidoreductases, including the members of the protein disulfide isomerase (PDI) family (Bechtel et al. ACS Chem Biol. 15, 543-553, 2020). In an oxidative environment of the in vitro experiments, it would be extremely difficult to prove that depalmitoylation promotes disulfide bond formation. It may be possible to validate this prediction using an assay system in which the whole process is performed under stringent (oxygen-free) nitrogen atmosphere so that the proteins do not have any contact with oxygen. The "Identification of PPT1 substrates highlight roles of depalmitoylation in disulfide bond formation and synaptic function" is of enormous importance in thiol biochemistry. However, it is critical that the authors provide solid evidence in support of this prediction.

Minor points-

1. Throughout the manuscript, numerous references are cited inappropriately. This should be corrected.

2. In Figure 1A the schematic of Acyl RAC assay may not be necessary as it is a widely used assay method (Forrester et al. J. Lipid Res. 2010).

Reviewer #3: 

The authors used a relatively straightforward (compared to ABE capture) method, Acyl-Resin Assisted capture for selectively enriching previously palmitoylated proteins and subsequent mass spectrometry identifications. The method relies on alkylation of all free thiols by NEM , followed by cleaving of palmitate moiety by hydroxylamine, exposing free thiols , which are then selectively captured using thiol reactive beads. Captured proteins are then analysed by tryptic digestion and label-free quantitative mass spectrometry.

The authors used this method in an attempt to identifying substrates of PPT1 using PPT1 KO mouse model. The assumption here was proteins with increased palmitoylation in KO, as compared to wild type mouse, would be the targets for the PPT1. In parallel, a global proteome quantification is also performed in order to distinguish palmitoylation increase from protein expression level increase. While global protein and palmitome levels did not significantly altered in whole brain, the same experiment on enriched synaptosome revealed several significant changes. Authors follow up this data with additional filtering and comparison with other databases in order to validate this list of potential substrates of PPT1. While this list of potential substrates can be a resource for future targeted validation, the authors did not mention some of the major limitations of this Acyl-RAC workflow.

Although a number of proteome -wide palmitoylation studies have been performed recently by employing Acyl-RAC based purifications, this method is prone to false positives due to hydrolysis of the thioester bond of other cysteine modifications, such as nitrosylation and glutathionylation. Moreover, the method cannot distinguish other lipid adducts on proteins which also form thioester bonds with cysteine and once cleaved, free thiols, thereby may be co-purified with thiol reactive beads. Additionally, thioesters are common in active site cysteine of many proteins and therefore may be co-purified even if not palmitoylated. Therefore the presence of carbamidomethylated peptide in the Acyl-RAC purified proteins, still provide only an indirect evidence of (previously) palmitoylated proteins.

In spite of these major limitations to the workflow, authors try to systematically and methodically refine the list of palmitoylated proteins. However, there appears some inconsistency in the proteome and palmitome coverage in various mass spec experiments. Firstly, overall proteome coverage (of 1873 common proteins) in mouse brain both genotypes, is relatively low for the type of instrumentation used. Secondly, while the first palmiotome screening of synapse identified 1378 proteins, the tertiary screening with dNEM identified in 3551 proteins in synapse and I wondered if there is any logical explanation for this. Probably protein sequence database search of MS data and preliminary filtering criteria have to be verified to prove these are genuine hits. Currently data (PXD017270) on PRIDE is not accessible.

Reviewer #4: This manuscript by Gorenberg et al describes very interesting new data regarding the potential substrates of palmitoyl protein thiosesterase 1. It has long been known that deficiency in this de-palmitoylating lysosomal hydrolase is the molecular cause of CLN1 disease, a fatal inherited and profoundly neurodegenerative disorder of childhood. However, progress in understanding the pathogenesis of this disorder has been severely hampered by not knowing the normal substrates that PPT1 acts upon. This manuscript utilizes a novel methodological strategy combining Acyl Resin-Assisted Capture and mass spectrometry to identify these substrates. The new strategy is based on identifying proteins with increased palmitoylation in PPT1 deficient mouse brains, and then because other depalmitoylating enzymes exist, validating these targets via recombinant PPT1. This has revealed evidence that about 10% of palmitoylated proteins at the synapse appear to be PPT1 substrates. The authors have sorted these into nine separate classes that are related to the phenotypes of PPT1 deficient mice and CLN1 patients. There is also evidence that the depalmitoylation sites are most often cysteine residues in disulfide bonds, suggesting a role for PPT1 and palmitoylation in regulating such interactions.

These studies appear to have been conducted rigorously and are presented very clearly so that they are relatively simple to understand, even for a reader who is not a specialist in proteomic analysis. The figures are especially well presented with good use of color coding to present several complicated data sets. Scientifically, the manuscript is of considerable importance in presenting significant novel data about PPT1 substrates that has been lacking for some time. This has been made possible by the application of a new method that has resolved a problem the field has been facing for a long time. In this respect it will be of considerable interest not just for those studying this and similar disorders, but also more widely for studying the importance of palmitoylation at the synapse in a range of disorders. Nevertheless, there remain a few, mostly conceptual issues that the authors should address in a revised manuscript.

a) The authors conducted their study in PPT1 mice at 2 months of age, but no rationale for chosing this age is given. This age is relatively early in disease progression, and is a sensible choice over later stages when many more downstream changes might be evident. Please can the authors explain their rationale?

b) The study was performed using whole brain extracts. These will necessarily contain neurons in addition to different populations of glia (astrocytes, microglia, oligodendrocytes). How can the authors account this mixed cell population or control for this? Or is this not a complicating factor for the conclusions they have reached?

c) Regionality is an important part of CLN1 pathogenesis, with markedly different onset and progression of pathology in the CLN1 brain and spinal cord. Have the authors considered including spinal cord samples in their analyses. Can the predict whether similar or different data may be produced?

d) Other depalmitoylating enzymes do not appear to be up regulated to compensate for lack of PPT1. Can the authors speculate further upon why this is the case? What does this tell us about the specificity of a depalmitoylating enzymes substrates?

e) From a neuroscience perspective, most of the synaptic targets appear to be pre-synaptic rather than post synaptic. Can the authors speculate further on the functional or mechanistic basis of this selectivity? What implications does this have for PPT1 function or CLN1 disease pathogenesis?

f) The authors predict that palmitoylated synaptic proteins traffic correctly in PPT1 mice, but may function poorly at the synapse due to absent or compromised depalmitoylation. This is a very interesting suggestion. please can the authors expand upon the rationale for this, and what further evidence would be needed to prove this hypothesis?

g) The higher power inserts of synaptic morphology in Figure 4D appear to lack a scale bar. Please can these be added.

---

## [Editor Report · Decision Letter 2]

13 Dec 2021

Dear Sreeganga,

Thank you again for submitting your manuscript "Identification of PPT1 substrates highlights roles of depalmitoylation in disulfide bond formation and synaptic function" for consideration as a Methods and Resources article at PLOS Biology. Your manuscript and rebuttal have been evaluated by the PLOS Biology editors and by the Academic Editor

In light of your response, we will welcome re-submission of a much-revised version that takes into account the reviewers' comments (previous decision letter). We cannot make any decision about publication until we have seen the final revised manuscript and your final response to the reviewers' comments. Your revised manuscript is also likely to be sent for further evaluation by the reviewers.

We expect to receive your revised manuscript within 3 months. 

**IMPORTANT - SUBMITTING YOUR REVISION**

*Re-submission Checklist*

*Published Peer Review*

*PLOS Data Policy*

*Blot and Gel Data Policy*

Sincerely,

Gabriel Gasque

Senior Editor

PLOS Biology

ggasque@plos.org

---

## [Editor Report · Decision Letter 3]

17 Feb 2022

Dear Sreeganga,

Thank you for submitting your revised Methods and Resources article entitled "Identification of PPT1 substrates highlights roles of depalmitoylation in disulfide bond formation and synaptic function" for publication in PLOS Biology. I have now discussed this new version with the Academic Editor, and I am pleased to tell you that we will probably accept this manuscript for publication, provided you satisfactorily address the following data and other policy-related requests:

1) Title: We would like to suggest a title that is more informative and appealing to a broad readership. We recommend: “Identification of substrates of palmitoyl protein thioesterase 1 highlights roles of depalmitoylation in disulfide bond formation and synaptic function.”

2) Ethics: Please include in your manuscript the ID number of the experimental protocol(s) approved by the Institutional Animal Care & Use Committee (IACUC) at Yale University.

3) Data: We note that you deposited the raw data into PRIDE. We could access PRIDE using the username and password provided but when we searched for the Project PXD017270, we could not find it. Could you please double check?

3.1) In addition, we must ask for all individual quantitative observations that underlie the data summarized in the figures and results of your paper. For an example see here: http://www.plosbiology.org/article/info%3Adoi%2F10.1371%2Fjournal.pbio.1001908#s5

These data can be made available in one of the following forms:

3.1.a) Supplementary files (e.g., excel). Please ensure that all data files are uploaded as 'Supporting Information' and are invariably referred to (in the manuscript, figure legends, and the Description field when uploading your files) using the following format verbatim: S1 Data, S2 Data, etc. Multiple panels of a single or even several figures can be included as multiple sheets in one excel file that is saved using exactly the following convention: S1_Data.xlsx (using an underscore).

3.1.b) Deposition in a publicly available repository. Please also provide the accession code or a reviewer link so that we may view your data before publication.

Regardless of the method selected, please ensure that you provide the individual numerical values that underlie the summary data displayed in the following figure panels: Figures 1D, 2BDEFH, 3E, 4DF, 5CDG, S1BDEFH, and S2C-E.

3.2) Please also ensure that each figure legend in your manuscript includes information on where the underlying data can be found and that your supplemental data file/s has/have a legend.

3.3) Please ensure that your Data Statement in the submission system accurately describes where your data can be found.

We expect to receive your revised manuscript within two weeks. 

*Published Peer Review History*

*Early Version*

Sincerely,

Gabriel

Gabriel Gasque, Ph.D.,

Senior Editor,

ggasque@plos.org,

PLOS Biology

---

## [Editor Report · Decision Letter 4]

2 Mar 2022

Dear Sreeganga,

On behalf of my colleagues and the Academic Editor, Gillian P. Bates, I am pleased to say that we can in principle accept your Methods and Resources "Identification of substrates of palmitoyl protein thioesterase 1 highlights roles of depalmitoylation in disulfide bond formation and synaptic function" for publication in PLOS Biology, provided you address any remaining formatting and reporting issues. These will be detailed in an email that will follow this letter and that you will usually receive within 2-3 business days, during which time no action is required from you. Please note that we will not be able to formally accept your manuscript and schedule it for publication until you have any requested changes.

***IMPORTANT: As you address these formatting and reporting issues please:

1) Ensure that each figure legend in your manuscript includes information on where the underlying data can be found (either PRIDE or S1 Data or both)

2) Ensure that your supplemental data file has a legend in the manuscript.

PRESS

Sincerely, 

Gabriel Gasque, Ph.D. 

Senior Editor 

PLOS Biology

ggasque@plos.org